# D-branes and orbit average

Peihe Yang[1⋆], Yunfeng Jiang[2,6†], Shota Komatsu[3,6‡] and Jun-Bao Wu[1,4,5∘]

**1** Center for Joint Quantum Studies and Department of Physics, School of Science,
Tianjin University, 135 Yaguan Road, Tianjin 300350, P. R. China
**2** Shing-Tung Yau Center and School of physics, Southeast University,
Nanjing 210096, China
**3** School of Natural Sciences, Institute for Advanced Study,
1 Einstein Dr. Princeton, NJ 08540, USA
**4** Peng Huangwu Center for Fundamental Theory, Hefei,
Anhui 230026, P. R. China
**5** Center for High Energy Physics, Peking University,
5 Yiheyuan Rd, Beijing 100871, P. R. China
**6** Department of Theoretical Physics, CERN,
1 Esplanade des Particules, 1211 Meyrin, Switzerland

⋆ peihe_yang@tju.edu.cn, ‡ shota.komatsu@cern.ch, ∘ junbao.wu@tju.edu.cn,
Corresponding author † Yunfeng.Jiang@cern.ch

The unusual ordering of authors instead of the standard alphabetical one in hep-th community is for students to get proper recognition of contribution under the current out-dated practice in China

## Abstract

We study correlation functions of D-branes and a supergravity mode in AdS, which are dual to structure constants of two sub-determinant operators with large charge and a BPS single-trace operator. Our approach is inspired by the large charge expansion of CFT and resolves puzzles and confusions in the literature on the holographic computation of correlation functions of heavy operators. In particular, we point out two important effects which are often missed in the literature; the first one is an average over classical configurations of the heavy state, which physically amounts to projecting the state to an eigenstate of quantum numbers. The second one is the contribution from wave functions of the heavy state. To demonstrate the power of the method, we first analyze the three-point functions in $\mathcal{N} = 4$ super Yang-Mills and reproduce the results in field theory from holography, including the cases for which the previous holographic computation gives incorrect answers. We then apply it to ABJM theory and make solid predictions at strong coupling. Finally we comment on possible applications to states dual to black holes and fuzzballs.

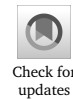

# 1   Introduction

In the top-down construction of AdS/CFT based on string theory, operators with different conformal dimensions admit different holographic descriptions. For instance in the matrix-like large $N$ limit, operators with $O(1)$ conformal dimension[1] are dual to perturbative string states while operators with $O(N)$ and $O(N^2)$ conformal dimensions correspond to D-branes and backreacted geometries including black holes. The best studied among them are operators

---

[1]More precisely what we mean here are operators whose dimensions do not scale with $N$.

dual to string states since they can be analyzed by various approaches such as supergravity, integrability, conformal bootstrap and perturbation theory. On the other hand, operators dual to black holes are least studied, yet most interesting since studying their correlation functions would allow us to address various important questions on quantum black holes. Eventually, we would like to understand operators dual to black holes but in this paper we set our goal more modest: We will study the correlation functions of operators, which are "in-between" string states and black-hole states—namely operators dual to D-branes. Only in the conclusion do we discuss applications to black holes.

This paper also serves as the second installment of our series of studies [1,2] on the structure constants of two determinant operators and a single-trace operator in ABJM theory. The main goal of this second paper is to analyze them at strong coupling using a dual description in terms of D-branes in AdS. However, the content of this paper is independent of the other two and it can be read separately.

To be concrete, we consider correlation functions of two (sub-)determinant operators, which are dual to D-branes called (non-)maximal giant gravitons, and a single-trace BPS operator. The holographic computation of such correlation functions was already performed in the literature both for $\mathcal{N} = 4$ super Yang-Mills (SYM) [3–6] and ABJM theory [7,8]. Unfortunately, there have been many puzzles and confusions and our main aim is to resolve them by presenting a streamlined analysis.

Let us first summarize what have been done in the literature.

1. The first attempts to compute these correlation functions were made in [3,4] for $\mathcal{N} = 4$ SYM. In particular in [4], they analyzed the *extremal* three-point functions and found that the results in the gauge theory and the holography look similar but do not quite match. This was rather puzzling since the structure constants of 1/2 BPS operators are known to be protected [9,10] and one would naively expect the two results to match perfectly. They then speculated that the mismatch is due to the inability of sub-determinant operators (also known as anti-symmetric Schur polynomial operators) to interpolate between a point-like graviton and a giant graviton. One basis that achieves the interpolation is the *single-particle basis* introduced originally by de Mello Koch and Gwyn in [11] and further studied in [12,13][2].

2. Subsequently, it was pointed out in [5] that the *non-extremal* three-point functions in $\mathcal{N} = 4$ SYM match perfectly between the gauge theory and the holography, for a special choice of a single-trace operator.

3. Analyses similar to the points 1 and 2 were made for ABJM theory in [7]. Here no definite conclusion was made since the three-point functions of BPS operators in ABJM theory are not protected and one cannot directly compare the results in the gauge theory and in the holography.

4. Later, it was realized in [6,8] that the holographic computation for the extremal three-point functions involves a (zero prefactor) × (divergent integral) structure. If one regularizes this quantity and takes a careful limit, it produces a finite correction which makes the final result match with the gauge theory answer. This regularization prescription was generalized and applied to ABJM theory in [8].

From this summary, one might get an impression that the problem was solved as long as one chooses a correct regularization prescription. However the "resolution" proposed in the literature is not satisfactory for several reasons:

---

[2]The name "single-particle basis" was used first in [13], which clarified various important properties of the basis including the vanishing of near extremal correlation functions.

- **Their regularization cannot be justified physically**: The three-point function studied in [8] is $\langle \chi_{J-k}(Z) \chi_J(\bar{Z}) \mathrm{tr}(Z^k) \rangle$ where $\chi_J$ is an anti-symmetric Schur polynomial of size $J$. To compute them, they first replaced the single-trace operator with $\mathrm{tr}(Z^k Y^l) + \cdots$ and then took the limit $l \to 0$. However once one modifies the operator to $\mathrm{tr}(Z^k Y^l) + \cdots$, the structure constant will vanish owing to the charge conservation. It is then rather puzzling that they got a finite answer and one could even suspect that this signals internal inconsistency of the computation (rather than providing a resolution of the mismatch). At a technical level, they obtained a non-zero answer because they performed the replacement only for the divergent integral and then added it back to other contributions which were computed using the un-modified operator. This is rather ad-hoc and hard to justify. Of course, one could possibly dismiss this as a minor concern which is unimportant as long as one gets the correct answer. However, as we see below, 小洞不補大洞吃苦[3].

- **It does not resolve all the mismatches**: The paper [5] studied a specific non-extremal three-point function and showed a match between the gauge theory and the holography. However, as we will show in this paper, for general non-extremal three-point functions of BPS operators (for which the regularization is not necessary), the holographic computation *does not* reproduce the result in the gauge theory if one simply follows the approach in [4]. This poses a sharper puzzle and cast doubt on the results for ABJM theory given in [7,8].

In this paper, we present a simple and streamlined analysis which resolves these puzzles and confusions. Our conclusion is simple to state:

> *The holographic computations performed in the literature are incomplete since they missed two important effects.*

Once these effects are taken into account, the holographic results for $\mathcal{N} = 4$ SYM match perfectly with the ones in the gauge theory, including the cases for which the previous approach fails to give the correct answers. The final results are given by highly nontrivial expressions involving the Legendre polynomials or the hypergeometric functions, and the precise match between field theory and holography gives us enough confidence on the validity of our approach. We then apply it to ABJM theory and make predictions for the structure constants at strong coupling. As a byproduct, we also draw the following conclusion:

> *As far as the CFT dual of giant gravitons is concerned, we did not find any evidence at strong coupling which favors the single-particle basis [11–13] over the more conventional Schur polynomial basis [14].*

Note that this is in contrast to point-like gravitons, for which there is a reason to prefer the single-particle basis as it has a direct connection to bulk vertices in supergravity [13]. For details, see section 4.4.

Let us now explain what these two effects are. The first and the most important effect is the *orbit average*, which was initially introduced for the structure constants of single-trace operators in [15]. Normally when one evaluates the three-point functions of two heavy operators and one light operator at strong coupling, one starts from a classical solution describing the two-point function of the heavy operators and perturbs it by the light operator. However the classical solution often comes with a moduli, namely there can be a family of solutions describing the same heavy operators. In such a case, one needs to perform an average over such classical solutions as was pointed out in [15]. Physically the average over the classical

---

[3]A Chinese saying meaning that *if one does not fix a small hole, one will suffer from a big hole later*.

solution converts a coherent state, which is a direct quantum analogue of a classical solution, to an eigenstate of quantum numbers such as the energy and the angular momenta. In this paper, we generalize the result of [15] to D-branes and show that the orbit average is crucial for reproducing the correct gauge theory answer. The second effect, which is important when the two heavy operators are not identical (we call such three-point functions *off-diagonal* in this paper), is the boundary term coming from the wave functions[4]. To see this, let us recall a typical extremal three-point function studied in the literature, $\langle \chi_{J-k}(Z) \chi_J(\bar{Z}) \text{tr}(Z^k) \rangle$. As is clear from this expression, the two heavy operators $\chi_{J-k}$ and $\chi_J$ are similar but not quite the same. In such cases, there is a nontrivial boundary contribution coming from a mismatch of the wave functions on the two ends of the classical solution. This gives a finite contribution which is needed to obtain the correct answer.

We should note that both of these effects are rather well-known in the context of the large charge expansion of CFTs [16] (see for instance the work by Monin, Pirtskhalava, Rattazzi and Seibold [17]). However, surprisingly the same analysis was never carried out in the current context. Obviously, the large charge expansion of CFTs shares much in common, both in philosophy and in techniques, with various concepts discussed in the integrability literature. We hope our work will help to bridge the knowledge gap in these two fields.

The rest of the paper is organized as follows: Before discussing the holographic computation of giant gravitons, we explain the basic idea in a simple quantum mechanical setup in section 2. After that we revisit the holographic computation of three-point functions of two non-maximal giant gravitons and a single-trace BPS operator in $\mathcal{N} = 4$ SYM. In section 3, we first focus on the *diagonal* three-point functions, namely the three-point functions for which two giant gravitons are identical up to complex conjugation, and discuss the necessity of performing the orbit average. We show that the result after the orbit average matches precisely with the result at weak coupling including the cases for which the previous approaches give wrong answers. We then proceed to discuss the *off-diagonal* three-point functions in $\mathcal{N} = 4$ SYM, namely the three-point functions for which the charges of two giant gravitons are not equal, in section 4. In this case, we show that there is an additional contribution coming from the wave functions. Once these effects are taken into account, the result coincides with the structure constant of two sub-determinant operators and a single-trace BPS operator in $\mathcal{N} = 4$ SYM. In section 5, we apply these new methods to compute the structure constants of two non-maximal giant gravitons and a single-trace BPS operator in ABJM theory and discuss the properties of the result. We then conclude and discuss future directions in section 6, including possible applications of our method to black holes, fuzzballs and superstrata. A few appendices are included to explain technical details.

## 2 A Toy Model

Let us first explain the two effects—the orbit average and the boundary terms from wave functions—in a simple quantum mechanical setup. This is essentially a review of the work by Monin, Pirtskhalava, Rattazzi and Seibold [17] (and partly [15]) but we highlight the importance of the two effects in the simplest possible setup and make some comments on how it applies to the holographic computation of giant gravitons.

---

[4]Similar effects were discussed in [15], but their analysis seems incomplete. In particular, their formulae do not reproduce the charge conservation, which we discuss in section 2.2.

## 2.1 Orbit average

Consider a quantum mechanical system with a $U(1)$ global symmetry, in which the degree of freedom lives on a circle $\theta \in [0, 2\pi]$ and the action $S[\theta]$ is invariant under the global $U(1)$ shift

$$\theta \to \theta + c\,. \tag{2.1}$$

We are interested in the expectation value of a light operator $\mathcal{O}$ for a state with a large $U(1)$ charge $|J\rangle$, namely $\langle J|\mathcal{O}(t=0)|J\rangle$, and evaluate it in the semi-classical (WKB) limit

$$J \to \infty\,, \qquad \hbar \to 0\,, \qquad \hbar J : \text{ fixed}\,. \tag{2.2}$$

In this limit, the wave function is given by the "WKB"-form,

$$\langle \theta|J\rangle = e^{iJ\theta}\,, \qquad \langle J|\theta\rangle = e^{-iJ\theta}\,, \tag{2.3}$$

and the path integral[5]

$$\langle J|\mathcal{O}(t=0)|J\rangle = \int D\theta(t)\, e^{-iJ\theta(t=+\epsilon)}\mathcal{O}[\theta(t=0)]e^{iJ\theta(t=-\epsilon)}e^{\frac{i}{\hbar}S[\theta]}\,, \tag{2.4}$$

can be evaluated by the stationary-phase, or equivalently saddle-point approximation. Here we shifted the insertion times of the wave functions $e^{\pm J\theta}$ by $\pm\epsilon$, but this is just for the convenience of explanation and the limit $\epsilon \to 0$ is usually non-singular.

The saddle-point in the WKB limit is given by

$$\frac{\delta S[\theta]}{\delta \theta(t)} + \hbar J \left(\delta(t+\epsilon) - \delta(t-\epsilon)\right) = 0\,. \tag{2.5}$$

Note that the operator $\mathcal{O}$ does not affect the saddle-point equation since we assumed that its quantum numbers are small (*i.e.* $\mathcal{O}$ is a light operator). Now, suppose we found one solution satisfying the equation (2.5), $\theta_0^*(t)$. Then, it immediately follows from the $U(1)$ invariance (2.1) that there should be a family of solutions, or equivalently a moduli of solutions, given by

$$\theta_c^*(t) \equiv \theta_0^*(t) + c\,, \qquad c \in [0, 2\pi]\,. \tag{2.6}$$

Therefore, the correct saddle-point formula is given by

$$\langle J|\mathcal{O}(t=0)|J\rangle \overset{\text{WKB}}{=} \int_0^{2\pi} \frac{dc}{2\pi}\, e^{-iJ\theta_c^*(+\epsilon)}\mathcal{O}[\theta_c^*(0)]e^{iJ\theta_c^*(-\epsilon)}e^{\frac{i}{\hbar}S[\theta_c^*]}\,. \tag{2.7}$$

In the limit $\epsilon \to 0$, the contributions from the two wave functions cancel. In addition, the action $S[\theta]$ is invariant under the shift by $c$ by assumption,

$$S[\theta_c^*] = S[\theta_0^*]\,. \tag{2.8}$$

Therefore we obtain a simpler expression

$$\langle J|\mathcal{O}(t=0)|J\rangle \overset{\text{WKB}}{=} e^{\frac{i}{\hbar}S[\theta_0^*]} \int_0^{2\pi} \frac{dc}{2\pi}\, \mathcal{O}[\theta_c^*(0)]\,. \tag{2.9}$$

As we can see, the final result is given by an average over the parameter $c$ and this is precisely the *orbit average* discussed in [15].

Note that the integral of $c$ is needed precisely because we wanted to evaluate the expectation value for the eigenstate of the $U(1)$ charge $|J\rangle$, which is invariant (up to a phase) under the $U(1)$ shift (2.1). If we instead used the coherent state, which is a direct quantum analogue of $\theta_0^*$, we would not need such averaging. To put it in another way, the orbit average is precisely what converts the expectation value for the coherent state into the expectation value for the $U(1)$ eigenstate.

---

[5]$\mathcal{O}[\theta]$ is given by $\langle\theta|\mathcal{O}|\theta\rangle$. We have assumed that $\langle\theta_1|\mathcal{O}|\theta_2\rangle$ is proportional to $\delta(\theta_1 - \theta_2)$.

## 2.2 Boundary term

Let us now generalize the computation slightly and consider the situation in which the bra and ket states are not identical: $\langle J + q | \mathcal{O} | J \rangle$. We assume $J$ is again large ($J \sim 1/\hbar \gg 1$) while $q$ is taken to be $O(1)$.

Following the aforementioned argument, we arrive at

$$\langle J + q | \mathcal{O}(t = 0) | J \rangle \overset{\text{WKB}}{=} \int_0^{2\pi} \frac{dc}{2\pi} e^{-i(J+q)\theta_c^*(+\epsilon)} \mathcal{O}[\theta_c^*(0)] e^{iJ\theta_c^*(-\epsilon)} e^{\frac{i}{\hbar}S[\theta_c^*]}. \tag{2.10}$$

The main difference from (2.7) is that now the contributions from wave functions do not cancel completely. Collecting the $c$-dependence, we arrive at the following formula:

$$\langle J + q | \mathcal{O}(t = 0) | J \rangle \overset{\text{WKB}}{=} e^{\frac{i}{\hbar}S[\theta_0^*]} \int_0^{2\pi} \frac{dc}{2\pi} e^{-iq\theta_c^*(0)} \mathcal{O}[\theta_c^*(0)]. \tag{2.11}$$

We can see that, as compared to (2.9), there is an extra factor $e^{-iqc}$ coming from the mismatch of the wave functions.

To see the physical significance of this extra factor, let us choose $\mathcal{O}$ to be the following simple operator with $U(1)$ charge $p$:

$$\mathcal{O}_p = e^{ip\theta}. \tag{2.12}$$

Substituting this expression into (2.11), we obtain

$$\langle J + q | \mathcal{O}_p(t = 0) | J \rangle \overset{\text{WKB}}{=} e^{\frac{i}{\hbar}S[\theta_0^*]} e^{i(p-q)\theta_0^*(0)} \int_0^{2\pi} \frac{dc}{2\pi} e^{i(p-q)c}. \tag{2.13}$$

Performing the integration over $c$, we then obtain

$$\langle J + q | \mathcal{O}_p(t = 0) | J \rangle \overset{\text{WKB}}{=} e^{\frac{i}{\hbar}S[\theta_0^*]} \delta_{p,q}. \tag{2.14}$$

Most notably, the final result contains a Kronecker delta $\delta_{p,q}$, which is a manifestation of the $U(1)$ charge conservation. This clearly demonstrates the necessity of the orbit average and the boundary term; if we did not take them into account, the final result would not obey the charge conservation—one of the fundamental properties of systems with global symmetry!

To summarize, the lessons that we can learn from this computation are

- First, when the bra and ket states are different, there is a nontrivial (boundary-term) contribution from the wave functions.

- Second, such contributions, together with the orbit average, are essential for reproducing a correct charge conservation $\delta_{p,q}$.

## 2.3 Orbit average and "symmetry breaking"

To apply the analyses in the previous subsections to giant gravitons, it is useful to restate the orbit average in terms of "symmetry breaking".

**Dimension of the moduli.** In the quantum mechanical toy model discussed above, the moduli of solutions was one dimensional since there was a single $U(1)$ symmetry. In general, if there are multiple commuting symmetries and the heavy states are eigenstates of all such symmetries, we would need to integrate over a multi-dimensional moduli space. This is particularly important if the system under consideration is integrable, as integrable theories have infinitely many commuting charges. However, it does not mean that we always need to integrate over an infinite dimensional moduli space for integrable theories. This is because the saddle-point solution ($\theta_0^*$) would be invariant under most of those infinite dimensional symmetries. So, more precisely, the dimensions of the moduli space $d_{\mathrm{mod}}$ is given by the following formula:

$$d_{\mathrm{mod}} = (\text{The number of commuting symmetries broken by the classical solution}). \quad (2.15)$$

Furthermore, in most cases, the right hand side of (2.15) is equal to

$$(\text{RHS of (2.15)}) = (\text{The number of nonzero charges of the heavy state}). \quad (2.16)$$

Therefore in practice the dimension of the moduli space is given by the number of non-vanishing (commuting) charges of the heavy state. We can see this also in the analysis of semi-classical string in [15, 18].

**Moduli average from orbit of broken symmetries.** Using this relation between the moduli and the broken symmetries, we can generate a family of classical solutions over which we perform averaging by simply acting broken symmetry generators to the original solution. In other words, the moduli of solutions can be identified with the orbit of the broken symmetry generators.

For the quantum mechanical setup discussed above, this is simply a change of viewpoints and does not affect the actual computation. However, this latter point of view is more advantageous when computing three-point functions of giant gravitons, and we will adopt it in the rest of this paper.

**Comments on the large charge expansion of CFT.** The discussions above might be reminiscent of the large charge effective field theory (EFT) of CFT [16, 17]. So let us clarify the precise relation between the two.

In the large charge EFT, we also count the number of symmetries broken by the classical solution. That gives the number of "Goldtstone bosons" which we use to write down the low-energy effective theory. However, we should keep in mind that the word "Goldstone bosons" is slightly abused here. Normally the Goldstone bosons are associated with a spontaneous symmetry breaking, which takes place only in the infinite volume limit. However, in the large charge expansion of CFT, we always consider a CFT defined on $R_t \times S^{d-1}$, which has a finite volume. As a consequence, the symmetry should never be spontaneously broken[6]. Nevertheless, as we said above, an individual semi-classical solution breaks some of the symmetries. What recovers the symmetries is precisely the integral over the moduli of solutions [16, 17], which is a space of zero modes of the "Goldstone bosons". Obviously the logic also applies to the quantum mechanical toy model discussed above.

This provides another argument for the necessity of the integration over the moduli; it is not a choice but something that is forced upon us in order to realize the symmetry structure of the problem correctly.

---

[6]Note that this is the case only for the internal symmetry. The spacetime symmetry such as translation and boost can be broken even in the finite volume. See [19] for discussions on the consequences of the boost symmetry breaking in conformal field theory.

## 2.4 Application to giant gravitons

Let us now briefly outline how the method applies to the three-point function of giant gravitons.

**Basic setup.** As already mentioned before, the main subject of this paper is the three-point functions of two sub-determinant operators and one single-trace BPS operator. In what follows the sub-determinant operator with charge $M$ will be denoted by $\mathcal{D}_M$ while the single-trace BPS operator with charge $L$ will be denoted by $\mathcal{O}_L$. In order to apply the argument in the previous subsections, we use the radial quantization and express the structure constant by the following matrix element;

$$\text{CFT}: \qquad C_{\mathcal{D}_{M+k}\mathcal{D}_M\mathcal{O}_L} = \langle \mathcal{D}_{M+k} | \mathcal{O}_L(t=0) | \mathcal{D}_M \rangle. \qquad (2.17)$$

Here we consider CFT defined on[7] $R_t \times S^{d-1}$, and $\langle \mathcal{D}_{M+k} |$ and $| \mathcal{D}_M \rangle$ are the bra and ket states corresponding to the operators $\mathcal{D}_{M+k}$ and $\mathcal{D}_M$ respectively. In particular, we are interested in the "heavy-heavy-light" three-point function satisfying $M \sim N \gg L, k$. As indicated, the operator $\mathcal{O}_L$ is inserted at $t = 0$.

To analyze (2.17) using holography, we simply need to replace each element on the right hand side with its holographic counterpart; or more precisely with quantities defined on the world-volume theory of the D-brane describing the giant gravitons. The counterparts of $\langle \mathcal{D}_{M+k} |$ and $| \mathcal{D}_M \rangle$ are quantum states of a giant graviton with angular momenta $M + k$ and $M$ defined on global AdS. On the other hand, $\mathcal{O}_L(t=0)$ is replaced by an operator defined on the world volume of the D-brane which describes a back reaction from the operator insertion $\mathcal{O}_L^\circ$ inserted at the boundary of AdS. As shown in [4–8], an explicit form of such an operator can be determined by perturbing the target space metric of the DBI action of the D-brane and it is given by a product of the bulk-to-boundary propagator and a spherical harmonics, integrated over the $t = 0$ slice of the D-brane worldvolume. For details, see sections 3 and 5. In summary, the expression for the structure constant is given by

$$\text{Holography}: \qquad C_{\mathcal{D}_{M+k}\mathcal{D}_M\mathcal{O}_L} = \langle \hat{\mathcal{D}}_{M+k} | \hat{\mathcal{O}}_L(t=0) | \hat{\mathcal{D}}_M \rangle, \qquad (2.18)$$

where we put hats to denote a holographic counterpart of each quantity. Here we choose a gauge in which the worldvolume time is identified with the time in global AdS, and denoted both by $t$.

**Semiclassical approximation.** The next step is to evaluate the right hand side of (2.18) by using the semiclassical approximation of the path integral of the worldvolume theory of the D-brane,

$$\langle \hat{\mathcal{D}}_{M+k} | \hat{\mathcal{O}}_L(t=0) | \hat{\mathcal{D}}_M \rangle = \int DX \, \Psi^*_{M+k}[X] \hat{\mathcal{O}}_L[X(t=0)] \Psi_M[X] e^{-S_{\text{DBI+WZ}}[X]}, \qquad (2.19)$$

where we denoted the fields on the worldvolume by $X$. In the semi-classical limit, this path integral is dominated by a classical solution satisfying the saddle-point equation. Let us denote one such solution by $X_0^*$. (The relevant solutions for $\mathcal{N} = 4$ SYM and ABJM theory were given in [4] and [7] respectively.) The giant gravitons discussed in this paper carry two non-vanishing charges, the conformal dimension $\Delta$ and the $U(1)$ R-charge $J$, and the classical solution $X_0^*$ breaks the corresponding two symmetries, the dilatation $D$ and the $U(1)$ R-charge rotation $\hat{J}$. Applying the arguments in the previous section, we can construct a two-parameter

---

[7]$S^3$ for $\mathcal{N} = 4$ SYM and $S^2$ for ABJM theory.

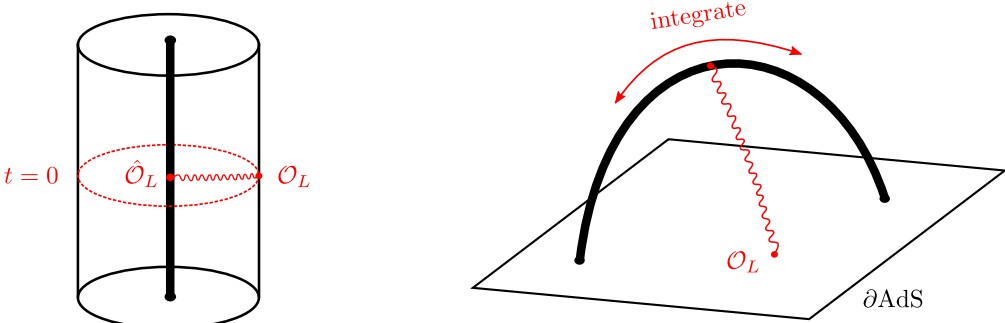

Figure 1: Comparison of our approach and the approach of [3, 4]. Left: In our approach, we use a radial-quantization picture and compute a matrix element of a light operator $\mathcal{O}_L$ between two D-brane states. This translates to a matrix element of a dual operator $\hat{\mathcal{O}}_L$ defined on the D-brane worldvolume. Right: The approach of [3, 4]. They considered a trajectory of a D-brane emitted and absorbed from the AdS boundary. They then attached a supergravity mode to the D-brane and integrated over its position.

family of solutions $X^*_{\tau_0,\phi_0}$ by acting these broken symmetry generators $e^{-D\tau_0}$ and $e^{i\hat{J}\phi_0}$ to $X^*_0$. In practice, these generators shift the corresponding target space coordinates. So we get

$$X^*_{\tau_0,\phi_0} = X^*_0\big|_{t\to t-i\tau_0,\,\phi\to\phi+\phi_0}\,. \tag{2.20}$$

Here $\phi$ is the target space coordinate conjugate to the $U(1)$ rotation $\hat{J}$ while $t$ is the global AdS time, which is conjugate to the dilatation[8]. Since the wave functions depend on these coordinates as $\Psi \sim e^{-i\Delta t + iJ\phi}$ ($\Psi^* \sim e^{i\Delta t - iJ\phi}$), these shifts result in the multiplication of the following factors to $\Psi$ and $\Psi^*$

$$\Psi \mapsto e^{-\Delta\tau_0} e^{iJ\phi_0}\Psi\,, \qquad \Psi^* \mapsto e^{+\Delta\tau_0} e^{-iJ\phi_0}\Psi^*\,. \tag{2.21}$$

Generalizing the argument for the quantum mechanical toy model, we then get the following semiclassical expression for $\langle\hat{\mathcal{D}}_{M+k}|\hat{\mathcal{O}}_L(t=0)|\hat{\mathcal{D}}_M\rangle$;

$$\langle\hat{\mathcal{D}}_{M+k}|\hat{\mathcal{O}}_L(t=0)|\hat{\mathcal{D}}_M\rangle \overset{\text{WKB}}{=} \underbrace{\int d\tau_0 \int \frac{d\phi_0}{2\pi}}_{\text{orbit average}} \hat{\mathcal{O}}_L[X^*_{\tau_0,\phi_0}(t=0)]\underbrace{e^{(\Delta_{M+k}-\Delta_M)\tau_0} e^{-i(J_{M+k}-J_M)\phi_0}}_{\text{wave function}}\,. \tag{2.22}$$

Here $\Delta_M$ and $J_M$ are the conformal dimension and the $R$-charge of the giant graviton with charge $M$. This is the master formula that we are going to use in the rest of this paper.

**Comparison with previous approaches.** Our master formula (2.22) differs in several ways from the expressions in [3–8], which are generalizations of the expression for the heavy-heavy-light three-point functions of string states proposed in [20, 21].

First, the papers [3–8] use a D-brane solution defined in the Poincaré AdS, which describes emission and absorption of a giant graviton from the AdS boundary (See Figure 1). This picture is more directly connected to the three-point function of CFT in $R^d$. On the other hand, we employed the radial quantization picture, which is more naturally related to global AdS, and considered a matrix element $\langle\mathcal{D}_{M+k}|\mathcal{O}_L|\mathcal{D}_M\rangle$ instead of the three-point function.

---

[8]The time evolution in global AdS is given by $e^{-iDt}$. As compared to the action of the symmetry generator $e^{-D\tau_0}$, it has an extra factor of $i$ and this is the reason for the imaginary shift $-i\tau_0$ in (2.20).

This latter picture makes the symmetries broken by the solution more manifest and therefore is advantageous for discussing the orbit average. It also makes it easier to write down the contributions from the wave functions.

Second, both (2.22) and the expressions in [3–8] contain an integral over the time variable $t$ or $\tau_0$, but the interpretations are quite different. In our formula, the $\tau_0$ integral comes from the orbit average, namely the average over classical solutions. Combined with the integration over the spatial worldvolume hidden in $\hat{\mathcal{O}}_L$, it reproduces an expression similar to the one in the papers [3–8]. On the other hand, the papers [3–8] consider a single classical solution. The integration over the time variable (and the spatial worldvolume) arises since the supergravity mode dual to the single-trace operator can hit any point on the worldvolume and one needs to integrate over all such possibilities.

Third, the integration over $\phi_0$ is completely lacking in the expressions in [3–8]. As discussed in the quantum mechanical toy model, this is necessary for realizing the correct charge conservation.

Finally, the papers [3–8] did not include the contributions from wave functions. Because of this, their results are insensitive to the details of the giant graviton states (namely whether the charges of the two giant gravitons are identical or not). Needless to say, the gauge theory answers do depend on such details and it is necessary to include such factors in order to reproduce the correct results.

# 3 Diagonal Three-Point Functions in $\mathcal{N} = 4$ SYM

In this section, we apply the method outlined in the previous section to compute the three-point function of two giant gravitons and a BPS single-trace operator in $\mathcal{N} = 4$ SYM. For simplicity, in this section we focus on the diagonal three-point function, for which the two giant gravitons have identical $R$ charges. The generalization to off-diagonal three-point functions will be discussed in the next section.

Before explaining the computation at strong coupling, let us clarify the setup in the gauge theory. We consider the anti-symmetric Schur polynomial operator $\chi_M(Z)$, which can be defined by a sub-determinant

$$\mathcal{D}_M = \chi_M(Z) \equiv \frac{1}{M!} \delta^{[b_1 b_2 \cdots b_M]}_{[a_1 a_2 \cdots a_M]} Z^{a_1}_{b_1} \cdots Z^{a_M}_{b_M}, \qquad \delta^{[b_1 \cdots b_M]}_{[a_1 \cdots a_M]} \equiv \sum_{\sigma \in S_M} (-1)^{|\sigma|} \delta^{b_1}_{a_{\sigma_1}} \cdots \delta^{b_M}_{a_{\sigma_M}}. \quad (3.1)$$

Here $Z$ is a complex scalar field of $\mathcal{N} = 4$ SYM. (For details of the notations and conventions used in this section, see [22].) To compute the structure constants of two such operators and a single-trace BPS operator, we consider the following matrix element

$$C_{\mathcal{D}_M \mathcal{D}_M \mathcal{O}_L} = \langle \mathcal{D}_M | \mathcal{O}_L | \mathcal{D}_M \rangle, \quad (3.2)$$

where $\mathcal{O}_L$ is defined by

$$\mathcal{O}_L \equiv \text{tr} \tilde{Z}^L, \qquad \tilde{Z} = \frac{Z + \bar{Z} + Y - \bar{Y}}{2}, \quad (3.3)$$

while $\langle \mathcal{D}_M |$ corresponds to $\chi_M(\bar{Z})$ inserted at infinity of $R^4$ and $|\mathcal{D}_M \rangle$ corresponds to $\chi_M(Z)$ inserted at the origin.

Thanks to supersymmetry, the structure constant (3.2) is independent of the coupling constant. Therefore the setup provides an ideal testing ground for our approach. In what follows, we perform the computation at strong coupling using a dual description of the D-brane, and show that the orbit average is necessary to reproduce the weak coupling result.

## 3.1 Structure constant from D3 brane

The anti-symmetric Schur polynomial operators are known to be dual to a D3-brane which is point-like in $AdS_5$ and extended in the $S^2$ subspace of $S^5$.

**D3 brane solution.**    To describe the solution, it is useful to express the metric of AdS$_5 \times S^5$ in terms of the global coordinates,

$$ds^2 = ds^2_{\text{AdS}} + ds^2_{S^5}, \tag{3.4}$$

where

$$ds^2_{\text{AdS}} = -\cosh^2\rho \, dt^2 + d\rho^2 + \sinh^2\rho \, d\widetilde{\Omega}_3^2,$$
$$ds^2_{S^5} = d\theta^2 + \sin^2\theta d\phi^2 + \cos^2\theta \, d\Omega_3^2, \tag{3.5}$$

where $d\widetilde{\Omega}_3^2$ and $d\Omega_3^2$ are the metric on $S^3$ which we parametrize as

$$d\widetilde{\Omega}_3^2 = d\tilde{\chi}_1^2 + \sin^2\tilde{\chi}_1 \, d\tilde{\chi}_2^2 + \cos^2\tilde{\chi}_1 d\tilde{\chi}_3^2,$$
$$d\Omega_3^2 = d\chi_1^2 + \sin^2\chi_1 \, d\chi_2^2 + \cos^2\chi_1 d\chi_3^2. \tag{3.6}$$

In terms of these coordinates, it is simple to write down a classical solution for the D3 brane: The solution is localized at $\theta = \theta_0$ and extended along $\chi_{1,2,3}$ directions. It is rotating along the $\phi$ direction at the speed of light. The worldvolume coordinates of the D3 brane $\sigma^\mu$ ($\mu = 0, 1, 2, 3$) are identified with the target space coordinates as follows:

$$\rho = 0, \qquad \sigma^0 = t, \quad \phi = t, \qquad \sigma^i = \chi_i, \quad i = 1, 2, 3. \tag{3.7}$$

The value of $\theta_0$ is related to the charge of the giant graviton as;

$$\cos^2\theta_0 = \frac{M}{N}, \tag{3.8}$$

where $\theta_0 = 0$ corresponds to the maximal giant graviton. Note that the classical D3-brane equations of motion lead to $\phi = t$.

In order to compute the structure constant using our master formula (2.22), we need to know $\hat{\mathcal{O}}_L$, which describes a small perturbation of the D3-brane action due to the backreaction from the supergravity mode. The D3-brane action consists of two terms, which are the DBI and the Wess-Zumino (WZ) actions. We will discuss them separately. Schematically, we have

$$\hat{\mathcal{O}}_L = \delta S_{\text{DBI}} + \delta S_{\text{WZ}}. \tag{3.9}$$

A few remarks are in order: First, although the analysis of a small perturbation is basically the same as what has been done in [4], there is one crucial difference. The paper [4] considered a perturbation on the whole Euclidean time domain $t_E \in [-\infty, \infty]$ while here we consider a perturbation localized at $t = 0$ slice since we are interested in the operator insertion $\hat{\mathcal{O}}_L$ at $t = 0$. Second, the right hand side of (3.9) is defined in terms of the D3-brane action in the Lorentzian signature. However, since we are interested in the $t = 0$ slice, which is shared with the Euclidean counterpart (namely the $t_E = 0$ slice), we can also use the Euclidean D3-brane action to read off the operator insertion. In this way we can recycle the results in [4].

**The DBI action.** The contribution from the DBI part is

$$\delta S_{\text{DBI}} = -\frac{N}{2\pi^2} \int d^3\sigma \ \delta\sqrt{h}\Big|_{t_E=0} , \tag{3.10}$$

where $h$ is the determinant of the induced metric on the worldvolume of the D3 brane. More explicitly, we have

$$\delta\sqrt{h} = \frac{1}{2}\sqrt{h}h^{ab}\delta h_{ab} = \frac{1}{2}\sqrt{h}h^{ab}\left(\frac{\partial X^\mu}{\partial\sigma^a}\frac{\partial X^\nu}{\partial\sigma^b}\delta g_{\mu\nu} + \frac{\partial X^\alpha}{\partial\sigma^a}\frac{\partial X^\beta}{\partial\sigma^b}\delta g_{\alpha\beta}\right), \tag{3.11}$$

where $\delta g_{\mu\nu}$ and $\delta g_{\alpha\beta}$ are the fluctuation of metrics on AdS and $S^5$ given by

$$\delta g_{\mu\nu} = \left[-\frac{5L}{6}g_{\mu\nu} + \frac{4}{L+1}\nabla_{(\mu}\nabla_{\nu)}\right]s^L(X)Y_L(\Omega), \tag{3.12}$$
$$\delta g_{\alpha\beta} = 2Lg_{\alpha\beta}s^L(X)Y_L(\Omega).$$

To proceed, we plug the proper spherical harmonics $Y_L(\Omega)$ and the bulk-to-boundary propagator $s^L(X)$ into (3.12). To write down the spherical harmonics $Y_L(\Omega)$ corresponding to $\text{tr}\tilde{Z}^L$, it is useful to use the embedding coordinates of $S^5$,

$$X\bar{X} + Y\bar{Y} + Z\bar{Z} = 1, \tag{3.13}$$

which are related to our coordinates as

$$X = \cos\theta\sin\chi_1 e^{i\chi_2}, \quad Y = \cos\theta\cos\chi_1 e^{i\chi_3}, \quad Z = \sin\theta e^{i\phi}. \tag{3.14}$$

Identifying these coordinates with scalar fields in $\mathcal{N} = 4$ SYM $(X, Y, Z)$ we find

$$Y_L(\Omega) = (\sin\theta\cos\phi + i\cos\theta\ \cos\chi_1\sin\chi_3)^L. \tag{3.15}$$

The embedding coordinates are also useful for writing down the bulk-to-boundary propagator $s^J(X)$ in the global AdS coordinates. The relation between the AdS embedding coordinates

$$-(X^0)^2 + (X^1)^2 + \cdots + (X^4)^2 + (X^5)^2 = -1, \tag{3.16}$$

and the global coordinates are given by

$$X^0 = \cosh t_E \cosh\rho, \quad X^\mu = n^\mu\sinh\rho, \quad X^5 = \sinh t_E\cosh\rho, \tag{3.17}$$

where $n^\mu$ is a unit vector. There is also a boundary version of the embedding coordinates[9] given by

$$P^0 = \cosh\bar{t}_E, \quad P^\mu = \bar{n}^\mu, \quad P^5 = \sinh\bar{t}_E, \tag{3.18}$$

and the bulk-to-boundary propagator is given by

$$s^L(X) \propto \frac{1}{(-2\ P|_{\bar{t}_E=0} \cdot X)^L}. \tag{3.19}$$

Here we set the time coordinate for $P$ to be zero since the boundary operator is inserted at $\bar{t}_E = 0$.

---

[9] Here we took the boundary of AdS to be $R \times S^3$.

As a result, we find

$$\delta S_{\text{DBI}} = -\frac{N}{2\pi}\cos^2\theta_0 \int_0^{2\pi} d\chi_3 \int_0^{\pi/2} d\chi_1 \, F_{\text{DBI}}\big|_{t_E=0} \, , \qquad (3.20)$$

where

$$F_{\text{DBI}} = \cos\chi_1 \sin\chi_1 \, Y_L(\Omega)\left(\frac{4}{L+1}\partial_{t_E}^2 - \frac{2L(L-1)}{L+1} - 8L\sin^2\theta_0 + 6L\right)s^L(t_E). \qquad (3.21)$$

Evaluating the bulk-to-boundary propagator (3.19) on the worldvolume of the D3-brane (namely setting $\rho = 0$) and including the normalization factor needed to unit-normalize the two-point function, we obtian

$$s^L(X) = \underbrace{\frac{1}{N}\frac{L+1}{4\sqrt{L}}}_{\text{normalization}}\left(\frac{1}{\cosh t_E}\right)^L . \qquad (3.22)$$

**The WZ action.** The contribution from the Wess-Zumino action is given by

$$\delta S_{\text{WZ}} = i\frac{N}{2\pi^2}\int d^3\sigma \, P[\delta C_E]\big|_{t_E=0} = \frac{4N}{2\pi^2}\int d^3\sigma \, \sqrt{g_{S^5}}s^L(t)\partial_\theta Y_L(\Omega)\big|_{t_E=0} . \qquad (3.23)$$

Plugging in the spherical harmonics and the bulk-to-boundary propagator, we arrive at

$$\delta S_{\text{WZ}} = \frac{N}{2\pi}\cos^2\theta_0 \int_0^{2\pi} d\chi_3 \int_0^{\pi/2} d\chi_1 \, F_{\text{WZ}}\big|_{t_E=0} \, , \qquad (3.24)$$

with

$$F_{\text{WZ}} = 8\cos\theta_0 \sin\theta_0 \cos\chi_1 \sin\chi_1 s^L(t)\partial_\theta Y_L(\Omega) . \qquad (3.25)$$

**Operator insertion.** Combining the two contributions, we find that the operator $\hat{\mathcal{O}}_L$ evaluated on the (unshifted) solution $X_0^*$ is given by

$$\hat{\mathcal{O}}[X_0^*] = -\frac{N}{2\pi}\cos^2\theta_0 \int_0^{2\pi} d\chi_3 \int_0^{\pi/2} d\chi_1 \, (F_{\text{DBI}} - F_{\text{WZ}})\big|_{t_E=0} . \qquad (3.26)$$

The expression for the shifted solution $X_{\tau_0,\phi_0}^*$ can be obtained by the replacements $t_E \to t_E + \tau_0$ and $\phi \to \phi + \phi_0$.[10] As a result, we find

$$\hat{\mathcal{O}}[X_{\tau_0,\phi_0}^*] = -\frac{N}{2\pi}\cos^2\theta_0 \int_0^{2\pi} d\chi_3 \int_0^{\pi/2} d\chi_1 \, [F_{\text{DBI}}(\tau_0,\phi_0) - F_{\text{WZ}}(\tau_0,\phi_0)] \, , \qquad (3.27)$$

with

$$\begin{aligned}
F_{\text{DBI}}(\tau_0,\phi_0) &= \frac{\sqrt{L}(L+1)}{2N}\left(\frac{\cos\phi_0\sin\theta_0 + i\cos\theta_0\cos\chi_1\sin\chi_3}{\cosh\tau_0}\right)^L \\
&\quad \times \sin(2\chi_1)\left(\cos(2\theta_0) + \tanh^2\tau_0\right), \\
F_{\text{WZ}}(\tau_0,\phi_0) &= \frac{\sqrt{L}(L+1)}{2N}\left(\frac{\cos\phi_0\sin\theta_0 + i\cos\theta_0\cos\chi_1\sin\chi_3}{\cosh\tau_0}\right)^L \\
&\quad \times \sin(2\theta_0)\sin(2\chi_1)\frac{\cos\phi_0\cos\theta_0 - i\sin\theta_0\cos\chi_1\sin\chi_3}{\cos\phi_0\sin\theta_0 + i\cos\theta_0\cos\chi_1\sin\chi_3} .
\end{aligned} \qquad (3.28)$$

---

[10]With the following orbit average in mind, we set $\phi = 0$ after this shifting. Precisely speaking, setting $t_E = 0$ in (3.20) and (3.23) should be done at this stage.

**Final result.** We can now plug (3.27) into our master formula (2.22),

$$C_{\mathcal{D}_M \mathcal{D}_M \mathcal{O}_L} = \int_{-\infty}^{\infty} d\tau_0 \int_0^{2\pi} \frac{d\phi_0}{2\pi} \hat{\mathcal{O}}_L[X^*_{\tau_0,\phi_0}]. \tag{3.29}$$

Note that the boundary terms from the wave functions cancel for the diagonal three-point functions. Evaluating the right hand side explicitly, we obtain the following closed-form expression (see Appendix B for the derivation):

$$C_{\mathcal{D}_M \mathcal{D}_M \mathcal{O}_L} = -\frac{i^L + (-i)^L}{2\sqrt{L}} \left( P_{\frac{L}{2}}(\cos 2\theta_0) + P_{\frac{L}{2}-1}(\cos 2\theta_0) \right). \tag{3.30}$$

Here $P_n$ is the $n$-th Legendre polynomial. As we will see below, this expression is in perfect agreement with the gauge theory results. Given the complexity of the final result (which contains Legendre polynomials), this match provides strong evidence for the validity of our approach.

**Comparison with previous approaches.** Let us perform a quick comparison with the previous approach. Precisely speaking, the three-point function that we analyzed was never studied in the literature but the results in [4] can be easily generalized to our case by replacing the spherical harmonics. The result of such a computation is given by the following integral, which takes a somewhat similar form to our final result:

$$C_{\mathcal{D}_M \mathcal{D}_M \mathcal{O}_L}^{[4]} = \int_{-\infty}^{\infty} d\tau_0 \, \mathcal{I}_L, \tag{3.31}$$

where the integrand $\mathcal{I}_L$ is given by

$$\mathcal{I}_L = -\frac{N}{2\pi} \cos^2 \theta_0 \int_0^{2\pi} d\chi_3 \int_0^{\pi/2} d\chi_1 \, (F_{\text{DBI}}^{[4]} - F_{\text{WZ}}^{[4]}), \tag{3.32}$$

with

$$
\begin{aligned}
F_{\text{DBI}}^{[4]} &= \frac{\sqrt{L(L+1)}}{2N} \left( \frac{\cosh\tau_0 \sin\theta_0 + i\cos\theta_0 \cos\chi_1 \sin\chi_3}{\cosh\tau_0} \right)^L \\
&\quad \times \sin(2\chi_1)\left(\cos(2\theta_0) + \tanh^2\tau_0\right), \\
F_{\text{WZ}}^{[4]} &= \frac{\sqrt{L(L+1)}}{2N} \left( \frac{\cosh\tau_0 \sin\theta_0 + i\cos\theta_0 \cos\chi_1 \sin\chi_3}{\cosh\tau_0} \right)^L \\
&\quad \times \sin(2\theta_0)\sin(2\chi_1)\frac{\cosh\tau_0 \cos\theta_0 - i\sin\theta_0 \cos\chi_1 \sin\chi_3}{\cosh\tau_0 \sin\theta_0 + i\cos\theta_0 \cos\chi_1 \sin\chi_3}.
\end{aligned}
\tag{3.33}
$$

Comparing (3.29) and (3.31), there are several important differences. First, the result obtained via the previous approach (3.31) does not contain integrals of $\phi_0$. Second, in the expressions of $F_{\text{DBI,WZ}}^{[4]}$, $\phi_0$ is replaced by $i\tau_0$. This is because we integrate over the whole Euclidean time domain in the approach of (3.31).

Due to these differences, the final results computed by (3.29) and (3.31) are different[11] for $\theta_0 \neq 0$. For comparison, we show the results for small values of $L$:

- $L = 2$

$$C_{\mathcal{D}_M \mathcal{D}_M \mathcal{O}_L} = \sqrt{2}(\cos\theta_0)^2, \qquad C_{\mathcal{D}_M \mathcal{D}_M \mathcal{O}_L}^{[4]} = \sqrt{2}(\cos\theta_0)^2[2 - \cos(2\theta_0)]. \tag{3.34}$$

---

[11]The results happen to agree for $\theta_0 = 0$, but this seems like a coincidence.

- $L = 4$

$$C_{\mathcal{D}_M \mathcal{D}_M \mathcal{O}_L} = \frac{1}{2}(\cos\theta_0)^2 [1 - 3\cos(2\theta_0)], \tag{3.35}$$

$$C^{[4]}_{\mathcal{D}_M \mathcal{D}_M \mathcal{O}_L} = \frac{1}{2}(\cos\theta_0)^2 [8 - 11\cos(2\theta_0) + \cos(4\theta_0)].$$

- $L = 6$

$$C_{\mathcal{D}_M \mathcal{D}_M \mathcal{O}_L} = \frac{1}{2\sqrt{6}}(\cos\theta_0)^2 [3 - 4\cos(2\theta_0) + 5\cos(4\theta_0)], \tag{3.36}$$

$$C^{[4]}_{\mathcal{D}_M \mathcal{D}_M \mathcal{O}_L} = \frac{1}{8\sqrt{6}}(\cos\theta_0)^2 [96 - 157\cos(2\theta_0) + 80\cos(4\theta_0) - 3\cos(6\theta_0)].$$

Since it is our result that agrees with the gauge theory answer, this shows the incompleteness of the previous approach.

## 3.2 Structure constant from gauge theory

We now compute the diagonal structure constant of two non-maximal giant gravitons and a single-trace BPS operator in $\mathcal{N} = 4$ SYM at weak coupling. As we will see below, the results match precisely with the holographic computations (3.30).

The computation in this subsection can be readily generalized to the dual giant gravitons as we show in Appendix A.

**Review of derivation of matrix product representation.** We use the effective field theory approach developed in [22–25] and derive a matrix product representation. The discussion below is mostly a review of those works and we refer to [22, 24] for more details.

We consider the generating function of giant gravitons,

$$\mathcal{G}_j \equiv \det[\mathbf{1} + t_j(Y_j \cdot \Phi)](x_j), \tag{3.37}$$

where $Y_j$ ($j = 1, 2$) is a six-dimensional null vector and $\Phi \equiv (\Phi_1, \ldots, \Phi_6)$ are the six real scalar fields in $\mathcal{N} = 4$ SYM. The giant graviton with a fixed charge $M$ can be obtained from the generating function by performing the integral over $t_j$:

$$\oint \frac{\mathrm{d}t_j}{2\pi i t_j^{1+M}} \mathcal{G}_j. \tag{3.38}$$

We then evaluate the correlation function of two generating functions and a BPS single-trace operator

$$\mathcal{O}_L(x_3) \equiv \mathrm{tr}\big((Y_3 \cdot \Phi)^L\big)(x_3), \tag{3.39}$$

at tree level. For a special choice of $Y_3$, this reduces to the operator (3.3) that we used in the holographic computation.

As the first step, we express the three-point function as a path integral

$$\langle \mathcal{G}_1 \mathcal{G}_2 \mathcal{O}_L \rangle = \frac{1}{Z_\Phi} \int D\Phi \left(\prod_{k=1}^{2} \mathcal{G}_k\right) \mathcal{O}_L \exp\left[-\frac{1}{g_{\mathrm{YM}}^2} \int \mathrm{d}^4 x \, \mathrm{tr}\big(\partial_\mu \Phi^I \partial^\mu \Phi_I\big)\right]. \tag{3.40}$$

Next we express the generating function in terms of integrals over fermions

$$\mathcal{G}_j = \int \mathrm{d}\bar{\chi}_j \mathrm{d}\chi_j \exp\big[\bar{\chi}_j(\mathbf{1} + t_j Y_j \cdot \Phi)\chi_j\big]. \tag{3.41}$$

We then integrate out scalar fields $\Phi^I$ to get an effective action for the fermions. As a result, we obtain the following expression

$$\langle \mathcal{G}_1 \mathcal{G}_2 \mathcal{O}_L \rangle = \int \left( \prod_{k=1}^{2} d\bar{\chi}_k d\chi_k \right) \mathcal{O}_L^S \exp \left[ \sum_{k=1}^{2} \bar{\chi}_k \chi_k - \frac{g^2}{N} \sum_{i \neq j} t_i t_j d_{ij} (\bar{\chi}_i \chi_j)(\bar{\chi}_j \chi_i) \right], \quad (3.42)$$

where $g^2 \equiv g_{YM}^2 N/(16\pi^2)$ and $d_{ij} \equiv Y_i \cdot Y_j / |x_{ij}|^2$, while $\mathcal{O}_L^S$ is given by

$$\mathcal{O}_L^S(x_3) \equiv \text{tr}\left( (Y_3 \cdot S)^L \right), \quad (3.43)$$

with

$$S^I \equiv \frac{g_{YM}^2}{8\pi^2} \sum_{k=1,2} \frac{t_k Y_k^I \chi_k \bar{\chi}_k}{|x_{k3}|^2}. \quad (3.44)$$

After that, we perform the Hubbard-Stratonovich transformation to rewrite the integral (3.42) into

$$\langle \mathcal{G}_1 \mathcal{G}_2 \mathcal{O}_L \rangle = \frac{1}{Z} \int d\rho d\bar{\chi} d\chi \, \mathcal{O}_L^S \exp \left[ \frac{2N}{g^2} \rho_{12} \rho_{21} + 2 \sum_{i \neq j} \hat{\rho}_{ij} (\bar{\chi}_j \chi_i) + \sum_{k=1}^{2} (\bar{\chi}_k \chi_k) \right], \quad (3.45)$$

with $\hat{\rho}_{ij} \equiv \sqrt{t_i t_j d_{ij}} \rho_{ij}$. Finally, We integrate out fermions and get

$$\langle \mathcal{G}_1 \mathcal{G}_2 \mathcal{O}_L \rangle = \frac{1}{Z} \int d\rho \, \left\langle \mathcal{O}_L^S \right\rangle_\chi \exp \left[ \underbrace{\frac{2N}{g^2} \rho_{12} \rho_{21} + N \log \left( 1 - 4 t_1 t_2 d_{12} \rho_{12} \rho_{21} \right)}_{\equiv S_{\text{eff}}} \right]. \quad (3.46)$$

Here $\left\langle \mathcal{O}_L^S \right\rangle_\chi$ can be computed by performing the Wick contractions of fermions[12]

$$\left\langle \bar{\chi}_i^a \chi_{j,b} \right\rangle = \delta_b^a (\Sigma^{-1})_{ij}, \quad (3.47)$$

with

$$\Sigma = \begin{pmatrix} 1 & 2\hat{\rho}_{12} \\ 2\hat{\rho}_{21} & 1 \end{pmatrix}. \quad (3.48)$$

In the large $N$ limit, the integral of $\rho$ in (3.46) can be approximated by the saddle point

$$\rho_{12}^* \rho_{21}^* = \frac{1}{4 t_1 t_2 d_{12}} - \frac{g^2}{2}, \quad (3.49)$$

and the saddle-point action is given by

$$S_{\text{eff}} = N \left( -1 + \frac{1}{2 g^2 t_1 t_2 d_{12}} + \log(2 g^2 t_1 t_2 d_{12}) \right). \quad (3.50)$$

Now, to compute the correlation functions of giant gravitons with fixed charges, we also need to perform integrals of $t_{1,2}$ (3.38). When the charge $M$ is of order $N$ (which is the case for

---

[12]Here $a$ and $b$ are indices for the $U(N)$ gauge group.

operators dual to D-branes), these integrals can also be evaluated at the saddle point. The saddle-point equation of these integrals is given by

$$t_1^* t_2^* = \frac{1}{2g^2 \sin^2 \theta_0 d_{12}} \, , \qquad (3.51)$$

where $\theta_0$ parametrizes the "non-maximality", namely

$$\frac{M}{N} \equiv \cos^2 \theta_0 \, . \qquad (3.52)$$

Note that this is the same parametrization as the holographic description of giant gravitons in (3.8).

At this point, let us make an important remark. As one can see from (3.51), the saddle point equation determines the product $t_1 t_2$ but not the ratio $t_1/t_2$. Therefore, when writing the expression for the three-point function, we still need to perform an integral over this ratio

$$y \equiv \sqrt{\frac{t_1}{t_2}} \, . \qquad (3.53)$$

Below we see how this modifies the matrix product representation.

As discussed in [22], the large $N$ limit also simplifies the computation of $\langle \mathcal{O}_L^S \rangle_\chi$ since the only contractions that survive in this limit are the ones that contract neighboring fermions. As a result, we obtain the following representation

$$\langle \mathcal{O}_L^S \rangle_\chi = - \oint_{|y|=1} \frac{\mathrm{d}y}{2\pi i y} \mathrm{Tr}\left[\mathcal{T}^L\right] , \qquad (3.54)$$

with

$$\mathcal{T} \equiv g \sqrt{\frac{2}{d_{12}}} \, \mathrm{diag}\left(d_{13} y, d_{23} y^{-1}\right) \cdot \begin{pmatrix} \sin \theta_0 & i \cos \theta_0 \\ i \cos \theta_0 & \sin \theta_0 \end{pmatrix} . \qquad (3.55)$$

Note that the integral $\oint_{|y|=1} \frac{\mathrm{d}y}{2\pi i y}$ comes from the integral over the ratio (3.53). The integral can be further simplified to

$$\langle \mathcal{O}_L^S \rangle_\chi = -(2g^2)^{\frac{L}{2}} \left(\frac{d_{13} d_{23}}{d_{12}}\right)^{\frac{L}{2}} \oint_{|y|=1} \frac{\mathrm{d}y}{2\pi i y} \mathrm{Tr}\left[\hat{\mathcal{T}}^L\right] , \qquad (3.56)$$

with

$$\hat{\mathcal{T}} \equiv \mathrm{diag}\left(y, y^{-1}\right) \cdot \begin{pmatrix} \sin \theta_0 & i \cos \theta_0 \\ i \cos \theta_0 & \sin \theta_0 \end{pmatrix} . \qquad (3.57)$$

**Evaluation of the matrix product.** Now, to evaluate (3.56), we first re-express it as

$$\oint_{|y|=1} \frac{\mathrm{d}y}{2\pi i y} \mathrm{Tr}\left[\hat{\mathcal{T}}^L\right] = \oint_{|s|=\epsilon \ll 1} \frac{\mathrm{d}s}{2\pi i s^{1+L}} \oint_{|y|=1} \frac{\mathrm{d}y}{2\pi i y} \mathrm{Tr}\left[\frac{1}{1-s\hat{\mathcal{T}}}\right] . \qquad (3.58)$$

The "generating function" $\frac{1}{1-s\hat{\mathcal{T}}}$ is expected to have a finite radius of convergence when expanded in powers of $s$. Therefore, we need to set $|s|$ sufficiently small to use the formula (3.58) and this is why the integration of $s$ is performed in a region $|s| \ll 1$. We then compute the right hand side of (3.58) by diagonalizing the matrix $1/(1-s\hat{\mathcal{T}})$. As a result, we get

$$\oint_{|y|=1} \frac{\mathrm{d}y}{2\pi i y} \mathrm{Tr}\left[\hat{\mathcal{T}}^L\right] = 2 \oint_{|s|=\epsilon \ll 1} \frac{\mathrm{d}s}{2\pi i s^{1+L}} \oint_{|y|=1} \frac{\mathrm{d}y}{2\pi i y} \frac{1 - \frac{s}{2}(y + \frac{1}{y}) \sin \theta_0}{1 - s(y + \frac{1}{y}) \sin \theta_0 + s^2} . \qquad (3.59)$$

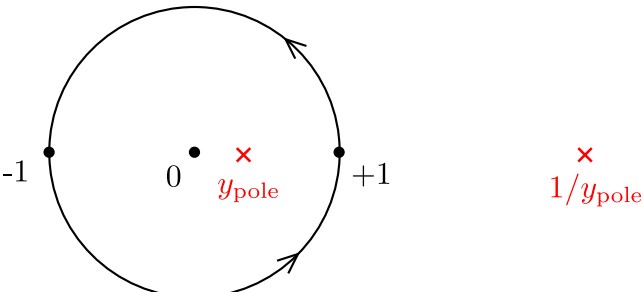

Figure 2: Positions of poles of the $y$-integral in (3.59). The integrand has two poles: for $s \ll 1$, one of them is inside the integration contour while the other is outside.

The next step is to perform the integration of $y$ by closing the contour and computing the residues of poles. Taking into account that $|s| \ll 1$, we find that there is one pole inside the contour

$$y_{\text{pole}} = \frac{1 + s^2 - \sqrt{1 + s^4 + 2s^2 \cos 2\theta_0}}{2s \sin \theta_0}.$$
(3.60)

It is straightforward to check that $|y_{\text{pole}}| < 1$ when $s$ is sufficiently small (see also Figure 2). Computing the residue, we obtain

$$\oint_{|y|=1} \frac{dy}{2\pi i y} \text{Tr}\left[\hat{\mathcal{T}}^L\right] = \oint_{|s|=\epsilon \ll 1} \frac{ds}{2\pi i s^{1+L}} \left[1 + \frac{1 - s^2}{\sqrt{1 + 2s^2 \cos 2\theta_0 + s^4}}\right].$$
(3.61)

Finally, the integral of $s$ can be performed using the formula for the Legendre polynomial $P_n$,

$$\frac{1}{\sqrt{1 - 2xt + t^2}} = \sum_{n=0}^{\infty} P_n(x) t^n.$$
(3.62)

Combining everything, we arrive at the final result

$$\left\langle \mathcal{O}_L^S \right\rangle_\chi = -(2g^2)^{\frac{L}{2}} \left(\frac{d_{13} d_{23}}{d_{12}}\right)^{\frac{L}{2}} \frac{i^L + (-i)^L}{2} \left(P_{\frac{L}{2}}(\cos 2\theta_0) + P_{\frac{L}{2}-1}(\cos 2\theta_0)\right),$$
(3.63)

which leads to the following result for the structure constant

$$C_{\mathcal{D}_M \mathcal{D}_M \mathcal{O}_L} = -\frac{i^L + (-i)^L}{2\sqrt{L}} \left(P_{\frac{L}{2}}(\cos 2\theta_0) + P_{\frac{L}{2}-1}(\cos 2\theta_0)\right).$$
(3.64)

This matches precisely with the result computed from holography (3.30).

# 4 Off-Diagonal Three-Point Functions in $\mathcal{N} = 4$ SYM

We now generalize the computations in the previous section to off-diagonal three-point functions, namely the three-point functions with two different giant gravitons.

## 4.1 Structure constant from D3 brane

Since we already determined the operator $\hat{\mathcal{O}}_L$ dual to the single-trace operator on the boundary, we simply need to use our master formula and include the contributions from the wave

functions. The result reads

$$C_{\mathcal{D}_{M+k}\mathcal{D}_M\mathcal{O}_L} = \int_{-\infty}^{\infty} d\tau_0 \int_0^{2\pi} \frac{d\phi_0}{2\pi} \hat{\mathcal{O}}_L[X^*_{\tau_0,\phi_0}] e^{k\tau_0} e^{-ik\phi_0}, \tag{4.1}$$

where $\hat{\mathcal{O}}_L[X^*_{\tau_0,\phi_0}]$ is given by the same expression as (3.27). Unlike the diagonal structure constant discussed in the previous section, we did not manage to derive a closed-form expression for general $k$, $M$ and $L$ analytically. However for given $k$, $M$ and $L$, the integral (4.1) can always be performed straightforwardly and we find that the results coincide with the following expression as long as $L$ is strictly larger than $k$:

$$C_{\mathcal{D}_{M+k}\mathcal{D}_M\mathcal{O}_L} =$$
$$-\frac{1}{2}\sqrt{L}\left(i^{L-k}+(-i)^{L-k}\right)\frac{\Gamma(\frac{L+k}{2})\cos^2\theta_0\sin^k\theta_0}{\Gamma(1+k)\Gamma(1+\frac{L-k}{2})} {}_2F_1\left(1+\frac{k-L}{2},1+\frac{k+L}{2},1+k;\sin^2\theta_0\right). \tag{4.2}$$

For $L < k$, the integral (4.1) simply vanishes, which is consistent with the $SO(6)$ selection rule. The only subtle case is $L = k$, which corresponds to the extremal three-point functions. In that case, the integral involves (zero prefactor) × (divergent integral) structure, much like what was observed in [6,8]. We will discuss this case in more detail in section 4.3.

We make a few comments before ending this subsection. First, let us emphasize that it is impossible to obtain a result like (4.2) with the approach of [4], since it is agnostic of the details of the giant gravitons. There is simply no room for including the $k$-dependence. Second, our result (4.2) is a highly nontrivial function of $\theta$ and $k$, which involves a hypergeometric function. In the next subsection, we will show that the same result can be obtained from $\mathcal{N}=4$ SYM. This provides another strong evidence supporting the validity of our method.

## 4.2  Structure constant from gauge theory

Let us now compute the off-diagonal structure constant from $\mathcal{N}=4$ SYM. In section 3.2, we computed the three-point functions of generating functions $\mathcal{G}_{1,2}$. Here we simply need to extract the off-diagonal structure constants by performing appropriate $t_{1,2}$ integrals. More precisely, we replace the $t_{1,2}$ integrals in (3.38) by

$$\oint \frac{dt_1}{2\pi i} \oint \frac{dt_2}{2\pi i} \frac{1}{(t_1 t_2)^{1+M}} \quad \mapsto \quad \oint \frac{dt_1}{2\pi i} \oint \frac{dt_2}{2\pi i} \frac{1}{(t_1 t_2)^{1+M+\frac{k}{2}}} \left(\sqrt{\frac{t_2}{t_1}}\right)^k. \tag{4.3}$$

Since $M \sim N \gg k$, we can approximate $(t_1 t_2)^{1+M+\frac{k}{2}}$ with $(t_1 t_2)^{1+M}$ at the leading order in the large $N$ expansion. This means that the saddle-point of the product $t_1 t_2$ (3.51) remains intact. The only modification is that we need to include a factor of

$$y^{-k} = \left(\sqrt{\frac{t_2}{t_1}}\right)^k, \tag{4.4}$$

when performing the integral (3.56). Namely we replace (3.56) with

$$\langle \mathcal{O}_L^S \rangle_\chi = -(2g^2)^{\frac{L}{2}} \left(\frac{d_{13}d_{23}}{d_{12}}\right)^{\frac{L}{2}} \oint_{|y|=1} \frac{dy}{2\pi i y^{1+k}} \text{Tr}[\hat{\mathcal{T}}^L]. \tag{4.5}$$

By performing the integral for various values of $L$ and $k$, we found that the result for the structure constant is given by

$$C_{\mathcal{D}_{M+k}\mathcal{D}_M\mathcal{O}_L} =$$
$$-\frac{1}{2}\sqrt{L}\left(i^{L-k}+(-i)^{L-k}\right)\frac{\Gamma(\frac{L+k}{2})\cos^2\theta_0\sin^k\theta_0}{\Gamma(1+k)\Gamma(1+\frac{L-k}{2})} {}_2F_1\left(1+\frac{k-L}{2},1+\frac{k+L}{2},1+k;\sin^2\theta_0\right), \tag{4.6}$$

which is in perfect agreement with the holographic result (4.2). One important difference from the holographic computation is that the integral (4.5) is well-defined even for the extremal case $L = k$ and gives (4.6).

For reader's convenience, let us also display the results for near-extremal three-point functions explicitly[13]:

$$k = L: \qquad C_{\mathcal{D}_{M+k}\mathcal{D}_M\mathcal{O}_k} = -\frac{\sin^k\theta_0}{\sqrt{k}}, \tag{4.7}$$

$$k + 2 = L: \qquad C_{\mathcal{D}_{M+k}\mathcal{D}_M\mathcal{O}_{k+2}} = \sqrt{k+2}\cos^2\theta_0\sin^k\theta_0, \tag{4.8}$$

$$k + 4 = L: \qquad C_{\mathcal{D}_{M+k}\mathcal{D}_M\mathcal{O}_{k+4}} = \frac{\sqrt{k+4}}{2}\cos^2\theta_0\sin^k\theta_0\left((3+k)\sin^2\theta_0 - (1+k)\right). \tag{4.9}$$

A couple of comments are in order: First, the result for the extremal structure constant (4.7) coincides with the gauge theory result in [4] computed by solving combinatorics of Wick contractions. This is of course as expected since both computed exactly the same three-point function.

Second, the results for the next to extremal case (4.8) and next-to-next extremal case (4.9) agree with the results in [13] obtained by the single-particle basis. This agreement is less trivial since the single-particle basis is different from sub-determinant operators (or equivalently the Schur polynomial basis), which we used here. More generally, the two bases seem to give the same answers in the large $N$ limit as long as the three-point function is non-extremal. Since these results are in agreement with the holographic results (4.2), this implies that, as long as non-extremal three-point functions are concerned, both the single-particle basis and the Schur polynomial basis are equally viable candidates for the holographic dual of the giant gravitons.

However, their results differ for the extremal three-point functions. As is shown in (4.7), the Schur polynomial basis gives $-\sin^k\theta_0/\sqrt{k}$ while the single-particle basis gives 0 (see [13]). This is simply because these two bases differ at the non-planar level and the extremal three-point function is sensitive to such difference. Now the question is which of these two bases is more naturally related to the holographic result. Our answer to this might be slightly disappointing since we will conclude that this question is ill-posed and we cannot provide a definite answer. See the following subsection for discussions on this point.

## 4.3 Extremal limit

We now discuss the extremal limit $k = L$ in more detail.

**Strong coupling.** As we mentioned in section 4.1, our integrand at strong coupling (4.1) contains a term of the form (zero prefactor) × (divergent integral) in the extremal limit. To see this, let us write down the integrand more explicitly:

$$C_{\mathcal{D}_{M+k}\mathcal{D}_M\mathcal{O}_L} = \int_{-\infty}^{\infty} d\tau_0\, e^{k\tau_0} \int_0^{2\pi} \frac{d\phi_0}{2\pi} e^{-ik\phi_0} \int_0^{2\pi} d\chi_3 \int_0^{\frac{\pi}{2}} d\chi_1 \left(\mathcal{I}_{\text{finite}} + \mathcal{I}_{\text{divergent}}\right), \tag{4.10}$$

where we split the integrand into a finite part and a divergent part (in the extremal limit):

$$\mathcal{I}_{\text{finite}} = \frac{4}{L+1}\cos\chi_1\sin\chi_1 Y_L(\Omega)(\partial_{t_E}^2 - L^2)s^L(t_E),$$

$$\mathcal{I}_{\text{divergent}} = 8\cos\chi_1\sin\chi_1 s^L(t_E)\underbrace{\left(L\cos^2\theta_0 - \cos\theta_0\sin\theta_0\partial_{\theta_0}\right)}_{\text{prefactor}}Y_L(\Omega). \tag{4.11}$$

---

[13]The first, second and third lines correspond to the extremal, next extremal and next-to-next extremal three-point functions respectively.

$\mathcal{I}_{\text{finite}}$ is well-defined and finite even in the extremal limit and gives

$$C^{\text{finite}}_{\mathcal{D}_{M+L}\mathcal{D}_M\mathcal{O}_L} = -\sqrt{L}(\cos\theta_0)^2(\sin\theta_0)^L. \tag{4.12}$$

On the other hand, the integral of $\mathcal{I}_{\text{divergent}}$ is divergent in the extremal limit while it contains a prefactor which vanishes in the limit. This signals a potential ambiguity in the final result coming from $0 \times \infty$. One way to properly address this term is to regulate the divergence and take a careful limit. The paper [8] proposed one such regularization (for a similar integral) which amounts to replacing the single-trace operator $\text{tr}(Z^k)$ with $\text{tr}(Z^k Y^l)$ and take the limit $l \to 0$. Unfortunately, as discussed in the introduction, their regularization is physically inconsistent.

A better way to deal with this problem is to first consider a non-extremal three-point function $L - k > 0$ and perform the analytic continuation to read off the result for the extremal limit $L - k = 0$. Our holographic result for the non-extremal three-point function is given by the analytic expression (4.2), which coincides with the gauge-theory result (4.6) even in the extremal limit. Therefore one might be tempted to conclude that our holographic computation correctly reproduces the gauge-theory result even in the extremal limit.

**Ambiguity in analytic continuation.** However there is one important caveat here: The procedure above requires us to analytically continue $L - k$ from positive integers. However the analytic continuation is not unique. For instance, we can add a term proportional to

$$\frac{\sin\pi(L-k)}{L-k}, \tag{4.13}$$

which vanishes for positive integer $L - k$ but changes the value at $L - k = 0$.

There are other situations in physics in which the analytic continuation from positive integers is required. However in most of such cases, there is some physical requirement which guarantees the uniqueness of the analytic continuation. For instance, in the computation of the entanglement entropy using the replica trick, one needs to analytically continue $n$ in $\text{Tr}[\rho^n]$, where $\rho$ is the density matrix. Since the eigenvalues of $\rho$ are all in the range $[0, 1]$, it satisfies the bound $|\text{Tr}[\rho^n]| \le \text{Tr}[\rho] = 1$ for $\text{Re } n > 1$. We can then apply Carlson's theorem to prove the uniqueness of the analytic continuation. Another situation in which the analytic continuation is required is the Regge theory of S-matrix [26] or CFT [27]. In those cases, we analytically continue spin which is originally taken to be positive integer. The uniqueness of the analytic continuation is guanrateed by the boundedness in the Regge limit and is manifested in the form of the Froissart-Gribov formula [26, 28].

Unfortunately, we do not know any such requirements in the present context and therefore cannot eliminate the ambiguity. This is perhaps not too surprising since similar ambiguities exist also in the target space analysis and in the gauge theory, as we see below.

**Boundary action in target space.** Similar puzzles and ambiguities surrounding the extremal limit are known also for the three-point functions of single-trace operators. Already in the early days of AdS/CFT, it was realized that the cubic coupling in type IIB supergravity in $AdS_5 \times S^5$ vanishes for the extremal configuration while the corresponding three-point function of single-trace operators is nonzero on the gauge-theory side. More generally, the bulk contact vertices are expected to vanish [29] for *near-extremal configurations*,

$$\langle s_{k_1} s_{k_2} \cdots s_{k_n}\rangle\big|_{\text{bulk}} = 0, \qquad \text{for } 0 \le -k_1 + \sum_{j=2}^{n} k_j \le 2(n-3), \tag{4.14}$$

where $s_k$ is a supergravity field dual to a chiral primary of charge $k$, while the corresponding correlation functions of single-trace operators on the gauge theory side do not vanish.

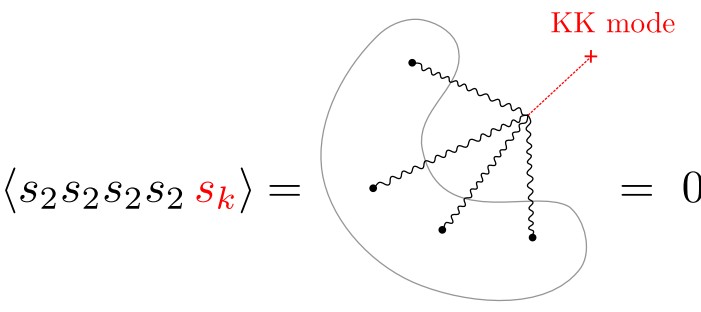

Figure 3: Relation between the vanishing of near-extremal correlation functions and the consistent truncation of supergravity. For $k_2 = \cdots = k_n = 2$, the condition (4.14) becomes equivalent to a physical requirement that the modes in the truncated supergravity do not source a higher Kaluza-Klein mode [29]. This is a necessary condition for the existence of the consistent truncation of type IIB supergravity on $AdS_5 \times S^5$.

For the special case of $k_2 = \cdots = k_n = 2$, this vanishing of the bulk vertices is required by the existence of the consistent truncation[14] of type-IIB supergravity in $AdS_5 \times S^5$ down to $\mathcal{N} = 8$ gauged supergravity in $AdS_5$ [29]. This is because $s_2$ corresponds to a field in the truncated supergravity and the equation (4.14) translates to the condition that the fields in the truncated supergravity ($s_2$) do not source $s_{k_1}$, which is in general a higher Kaluza-Klein mode (*i.e.* a mode excluded in the truncated theory). See also Figure 3. Thus in summary, non-zero answers for these near-extremal correlation functions on the gauge-theory side seem to be in conflict with the consistent truncation of supergravity which we know to exist.

For three-point functions, a resolution of this puzzle was found in [30]. They pointed out that the bulk integral for the Witten diagram diverges while the overall prefactor tends to zero as we take the extremal limit. Therefore, one again faces the "$0 \times \infty$ problem". In order to remedy this, they introduced a cut-off in the radial direction and carefully analyzed the boundary term in the supergravity action. This leads to a finite result which matches precisely the gauge theory answer for single-trace operators.

Such boundary interactions[15] in the target space would be important also for the Giant Graviton. However it is hard to take them into account in the current analysis since they correspond to emission and absorption of supergravity modes that only take place at the boundary of AdS and cannot be seen in the standard DBI and WZ actions of the D-branes.

To make things worse, the boundary terms of the supergravity action may not be unique. In some cases such as $AdS_4 \times S^7$ discussed in [31], the boundary term is determined uniquely by the requirement of supersymmetry. However, this does not seem to be the case in the current context. For instance, a recent paper [13] proposed a different basis (single-particle basis) on the gauge-theory side which is a mixture of single-trace and multi-trace operators. With this new basis, they showed that all the near extremal correlation functions vanish being consistent with the structure of the supergravity vertices (4.14). However as shown in [30], there is a way to perform the computation in supergravity so that it leads to finite answer. This does not mean that there is a conflict between [30] and [13]. It simply means that the basis defined in [13] would correspond to a different choice of the boundary term in the supergravity action. Such extra boundary terms are known to arise from field definitions in the bulk as was shown

---

[14]SK thanks Alexandre Belin and Nikolay Bobev for discussions on related topics.

[15]We should not confuse this with the boundary term coming from wave functions discussed in this paper. The latter is a boundary term on the worldvolume of the giant graviton while here we are talking about the boundary term in the target space effective field theory.

by Arutyunov and Frolov [32][16]. So, without specifying the boundary terms (or equivalently without giving a detailed definition of the fields in the bulk), even a question of which operator is dual to a given mode in supergravity becomes ill-posed. Needless to say, only after settling that question can we compare the results in the gauge theory and the supergravity.

**Multi-trace mixing in $\mathcal{N} = 4$ SYM.** Let us also point out that the ambiguity related to the extremal three-point function is present also on the gauge theory side. As discussed in [30], the (near-)extremal correlation functions are sensitive to how the operators are defined at the non-planar level. Normally adding the multi-trace terms to the single-trace operator does not modify the large $N$ three-point function as long as the coefficients of the multi-trace terms are $1/N$ suppressed[17]. However, as discussed in [30], the contributions from the multi-trace terms get enhanced for the extremal three-point functions. Therefore, such an addition of multi-trace operators allows us to modify the extremal three-point functions without modifying the non-extremal correlation functions. This ambiguity is the gauge theory counterpart of the ambiguity related to the boundary term of the supergravity action discussed above.

**Summary.** Given all these ambiguities, we feel that comparing the extremal three-point functions between the gauge theory and the supergravity is an ill-posed question. This may be a somewhat disappointing conclusion but we want to emphasize the following two points: First, all these ambiguities are absent in non-extremal three-point functions. Therefore, non-extremal three-point functions provide an ideal testing ground for the holographic computation and our results pass that test in a highly nontrivial manner. Second, the paper [4], which computed the extremal three-point functions of giant gravitons, speculated that the mismatch they found may be due to inability of Schur polynomials to interpolate between a point-like graviton and a giant graviton. This claim seems false since the Schur polynomial basis works perfectly for the non-extremal three-point functions (including the three-point functions of non-maximal giant gravitons[18]). Instead the "mismatch" is due to the ambiguities inherent in the extremal three-point functions, which we discussed at length in this subsection.

## 4.4 Single-particle basis vs. Schur polynomial

With the discussions above in mind, we now compare the single-particle basis and the Schur-polynomial basis as potential CFT duals of giant gravitons. For this purpose, let us first summarize a couple of facts:

- For non-extremal three-point functions, the single-particle basis and the Schur polynomial basis gives identical results and they agree with the holographic computations.

- For the extremal three-point functions, the Schur-polynomial basis predicts (4.6) while the single-particle basis predicts 0.

- On the holographic side, the "simplest" way to compute the extremal three-point function is to set the $(0 \times \infty)$ factor to zero. An analog of this for point-like supergravity modes is to focus on the bulk vertices in supergravity and discard other contributions, which gives a result that agrees with the single-particle basis. However, for giant gravitions, this procedure gives (4.12), which does not match either with the single-particle basis or with the Schur basis.

---

[16]We thank Francesco Aprile, James Drummond, Paul Heslop and Michele Santagata for pointing this out and patiently explaining it to us.

[17]This suppression indeed happens when we diagonalize the two-point functions [30].

[18]Moreover, as we discussed below (4.7)–(4.9), the Schur polynomial basis gives the same non-extremal three-point functions as the single-particle basis, which interpolates between a single-trace operator and a determinant operator [13].

- The "second simplest" way to compute it is to first consider the non-extremal three-point function and then analytically continue. One natural analytic continuation gives the result consistent with the Schur basis (but not with the single-particle basis).

- However, since the holographic computation is subject to various ambiguities discussed above, one cannot draw a definite conclusion.

- It is likely that the two bases simply correspond to different choices of the boundary terms in AdS and that they are related by the field redefinition in the bulk[19] [32].

We therefore conclude that, as a CFT dual of the giant graviton, there is no reason to favor one basis over the other. We emphasize that the logic that worked for point-like supergravity modes and favored the single-particle basis does not work for giant gravitons since there is no "simple" computation on the holographic side that reproduces the result in the single-particle basis.

## 5 Application to ABJM Theory

In this section, we apply the prescriptions developed in the previous sections to the computation of three-point functions involving two giant gravitons and one single-trace BPS operator in ABJM theory. For simplicity, we focus on the diagonal case and take two anti-symmetric Schur polynomials with the same $R$-charge, but the generalization to off-diagonal three-point functions is straightforward. The gravity dual of these operators are D4-branes which are point like in AdS and extended in $CP^3$ directions. Before we move to the strong coupling computation, let us first review our setup at weak coupling which has been discussed in the first paper [1]. Notice that in ABJM theory, these three-point functions are no longer protected by supersymmetry and we do not expect a match between gauge theory and holography.

Apart from not being protected, there is another important difference between the structure constants of BPS operators in ABJM theory and $\mathcal{N} = 4$ SYM. The $R$-symmetry structure of the BPS three-point functions is completely fixed in $\mathcal{N} = 4$ SYM, while in ABJM theory we have different structures. For the three-point function under consideration, we have the following structure [1]

$$\frac{\langle \mathcal{D}_M(x_1; n_1, \bar{n}_1) \mathcal{D}_M(x_2; n_2, \bar{n}_2) \mathcal{O}_L(x_3; n_3, \bar{n}_3) \rangle}{\mathcal{N}_{\mathcal{D}_M} \sqrt{\mathcal{N}_{\mathcal{O}_L}}} = (d_{12} d_{21})^M \left( \frac{d_{23} d_{32} d_{31} d_{13}}{d_{12} d_{21}} \right)^{\frac{L}{2}} \sum_{p=-\frac{L}{2}}^{\frac{L}{2}} D_{M|L}^{(p)} \xi^p,$$

where $D_{M|L}^{(p)}$'s are structure constants and $\xi$ is the $R$-symmetry cross ratio defined by

$$\xi \equiv \frac{(n_1 \cdot \bar{n}_2)(n_2 \cdot \bar{n}_3)(n_3 \cdot \bar{n}_1)}{(n_2 \cdot \bar{n}_1)(n_3 \cdot \bar{n}_2)(n_1 \cdot \bar{n}_3)}, \tag{5.1}$$

$d_{ij}$'s are defined by

$$d_{ij} \equiv \frac{n_i \cdot \bar{n}_j}{|x_{ij}|}, \qquad |x_{ij}| \equiv |x_i - x_j|, \tag{5.2}$$

and $\mathcal{N}_{\mathcal{D}_M}, \mathcal{N}_{\mathcal{O}_L}$ are defined by

$$\langle \mathcal{D}_M(x_1, n_1, \bar{n}_1) \mathcal{D}_M(x_2, n_2, \bar{n}_2) \rangle = \mathcal{N}_{\mathcal{D}_M} (d_{12} d_{21})^M, \tag{5.3}$$

---

[19]We thank Francesco Aprile and Paul Heslop for emphasizing this point.

and

$$\langle \mathcal{O}_{L_1} \mathcal{O}_{L_2} \rangle = \delta_{L_1, L_2} \mathcal{N}_{\mathcal{O}_{L_1}} (d_{12} d_{21})^{L_1}, \tag{5.4}$$

respectively.

This implies for generic polarizations of the three operators, we need to compute $L + 1$ structure constants for the single trace operator of length $L$. We will focus on a special choice of the polarization such that $\xi = -1$ in the main text while relegating the discussions on more general kinematics to Appendix D. The choice $\xi = -1$ corresponds to considering the three operators in the so-called twisted translated frame. See [1] for more details. What we are actually computing in this case is the following structure constant

$$C_{\mathcal{D}_M \mathcal{D}_M \mathcal{O}_L} = \sum_{p=-\frac{L}{2}}^{\frac{L}{2}} (-1)^p D_{M|L}^{(p)}. \tag{5.5}$$

More specifically, we consider the following sub-determinant operators

$$\mathcal{D}_M(x; n_j, \bar{n}_j) \equiv \frac{1}{M!} \delta_{[a_1 \cdots a_M]}^{[b_1 \cdots b_M]} \big[ (n_j \cdot Y)(\bar{n}_j \cdot \bar{Y}) \big]_{b_1}^{a_1} \cdots \big[ (n \cdot Y)(\bar{n} \cdot \bar{Y}) \big]_{b_M}^{a_M}, \qquad j = 1, 2. \tag{5.6}$$

The polarizations are

$$\begin{aligned} n_1 &= (1, 0, 0, 0), & \bar{n}_1 &= (0, 0, 0, 1), \\ n_2 &= (0, 0, 0, 1), & \bar{n}_2 &= (1, 0, 0, 0). \end{aligned} \tag{5.7}$$

As for the single-trace operator, we choose the polarization vectors to be

$$n_3 = \frac{1}{\sqrt{2}} (1, 0, 0, -1), \qquad \bar{n}_3 = \frac{1}{\sqrt{2}} (1, 0, 0, 1). \tag{5.8}$$

Therefore, the single trace operator of length $2L$ reads

$$\mathcal{O}_L = \frac{1}{2^L} \mathrm{tr} \Big[ \big( (Y^1 - Y^4)(\bar{Y}_1 + \bar{Y}_4) \big)^L \Big]. \tag{5.9}$$

## 5.1 Structure constants from D-brane

We now compute the structure constants using D-branes. For this purpose, we follow the approach of [8] and consider first a classical solution of the M5-brane in $AdS_4 \times S^7 / Z_k$. Although this is not the limit we are interested in (since we are studying the type IIA limit), we can still perform the computation using this solution and later take the limit $k \to \infty$ to dimensionally reduce the target space to $AdS_4 \times CP^3$. After taking this limit, the M5-brane becomes a D4-brane in type-IIA supergravity.

**M5 brane solution.** Let us write down the M5 brane solution that corresponds to the antisymmetric Schur polynomial. The metric reads

$$ds^2 = R_{\mathrm{AdS}}^2 \, ds_{\mathrm{AdS}}^2 + R_{S^7}^2 \, ds_{S^7/Z_k}^2, \tag{5.10}$$

where $R_{S^7} = 2 R_{\mathrm{AdS}}$ and

$$R_{S^7} = (32 \pi^2 k N)^{\frac{1}{6}} \ell_P. \tag{5.11}$$

Here $\ell_P$ is related to the M5-brane tension by $T_{\mathrm{M5}}^{-1} = (2\pi)^5 \ell_P^6$.

The $\text{AdS}_4$ metric in the global coordinates reads

$$ds^2_{\text{AdS}} = -\cosh^2 \mu \, dt^2 + d\mu^2 + \sinh^2 \mu \, d\Omega_2^2, \tag{5.12}$$

where $d\Omega_2^2$ is the metric on the 2-sphere

$$d\Omega_2^2 = d\tilde{\theta}^2 + \sin^2 \tilde{\theta} \, d\tilde{\varphi}^2. \tag{5.13}$$

For the metric on $S^7/Z_k$, it is convenient to use the following parametrization[20]

$$
\begin{aligned}
Z_1 &= r \, \exp[\rho + i(\chi/2 + \phi + \theta)], \\
Z_2 &= \tilde{r} \, \exp[i\phi], \\
Z_3 &= \exp[\rho_3 + i(\theta_3 + \phi)], \\
Z_4 &= r \, \exp[-\rho + i(-\chi/2 + \phi + \theta)],
\end{aligned}
\tag{5.14}
$$

where $\tilde{r}$ is given by

$$\tilde{r}^2 = 1 - 2r^2 \cosh(2\rho) - e^{2\rho_3}. \tag{5.15}$$

These coordinates are in one-to-one correspondence to the four scalar fields $Y^I$ in ABJM theory, namely $Z_I \leftrightarrow Y^I$, $\bar{Z}_I \leftrightarrow \bar{Y}_I$, ($I = 1, 2, 3, 4$).

The coordinates (5.14) cover the unit $S^7$. We have

$$|Z_1|^2 + |Z_2|^2 + |Z_3|^2 + |Z_4|^2 = 1. \tag{5.16}$$

The range of the 'radial' coordinates $r, \rho_3, \rho$ are

$$0 \leq \rho_3 \leq \rho_3^{\max}(r, \rho), \qquad -\rho^{\max}(r) \leq \rho \leq \rho^{\max}(r), \qquad 0 \leq r \leq 1/\sqrt{2}, \tag{5.17}$$

where $\rho_3^{\max}(r, \rho)$ and $\rho^{\max}(r)$ are given by the following equations

$$e^{2\rho_3^{\max}(r,\rho)} = 1 - 2r^2 \cosh(2\rho), \qquad \cosh[2\rho^{\max}(r)] = \frac{1}{2r^2}. \tag{5.18}$$

The range of the angular coordinates are

$$0 \leq \theta, \theta_3, \chi, \phi \leq 2\pi. \tag{5.19}$$

Defining $z_i = Z_i/Z_4$ ($i = 1, 2, 3$), The $S^7$ metric can be written as

$$ds^2_{S^7} = \frac{dz_i d\bar{z}_j}{(1 + z_k \bar{z}_k)^2} [\delta_{ij}(1 + z_k \bar{z}_k) - \bar{z}_i z_j] + (d\phi + A)^2, \tag{5.20}$$

where $A$ is the 1-form

$$A = \frac{i}{2(1 + z_k \bar{z}_k)} (z_i d\bar{z}_i - \bar{z}_i dz_i). \tag{5.21}$$

Repeated indices are summed over. The quotient space $S^7/Z_k$ is obtained by restricting the range of the angle $\phi$ to $0 \leq \phi \leq 2\pi/k$.

A classical solution for the M5 brane is given by the curve

$$Z_1 \bar{Z}_4 = \alpha^2 \, e^{it} \tag{5.22}$$

---

[20]We swapped the definition of $Z_2$ and $Z_4$ compared to [7] in accordance with our gauge theory convention.

in $S^7/Z_k$. The parameter $\alpha$ determines the $R$-charge of the M5-brane solution and is related to the charge of the sub-determinant operator in the following way [7,33]:

$$\frac{M}{N} = \sqrt{1-4\alpha^4} - 4\alpha^4 \log\left(\frac{1+\sqrt{1-4\alpha^4}}{2\alpha^2}\right). \tag{5.23}$$

In terms of the coordinates introduced above (see for instance (5.14)), the curve (5.22) corresponds to setting $\mu = 0, r = \alpha$ and $\chi = t$. The identification $\chi = t$ comes from the classical equations of motion of the M5-brane [7,33]. The rest of the coordinates $(t, \rho, \rho_3, \theta, \theta_3, \phi)$ are identified with the coordinates of the world volume of the M5-brane and they vary in the ranges

$$-\rho^{\max} \le \rho \le \rho^{\max}, \qquad 0 \le e^{2\rho_3} \le e^{2\rho_3^{\max}(\rho)}, \qquad 0 \le \theta, \theta_3, \phi \le 2\pi, \tag{5.24}$$

with

$$e^{2\rho_3^{\max}(\rho)} = 1 - 2\alpha^2 \cosh(2\rho), \qquad \cosh(2\rho^{\max}) = \frac{1}{2\alpha^2}. \tag{5.25}$$

**The DBI action.** The operator insertion corresponding to the fluctuation of the M5-brane is the sum of the DBI action and the WZ action. The DBI action takes the following form

$$\delta S_{DBI} = -\frac{R_{AdS}^6}{(2\pi)^5 \ell_P^6} \frac{(2\pi)^3}{k} \int_{-\rho^{\max}}^{+\rho^{\max}} d\rho \, F_{DBI}\big|_{t_E=0}, \tag{5.26}$$

where

$$F_{DBI} = 2^4 \alpha^2 Y_L(\Omega)\left[1 - 2\alpha^2 \cosh(2\rho)\right]\left[\cosh(2\rho) - 2\alpha^2\right] \tag{5.27}$$
$$\times \left(4L + \frac{\cosh(2\rho)}{\cosh(2\rho) - 2\alpha^2}\left[\frac{2}{L+1}\partial_{t_E}^2 - \frac{2L^2}{L+1}\right]\right)s^L(t_E).$$

The spherical harmonics corresponding to the single trace operator (5.9) is given by

$$\mathcal{O}_J \mapsto Y_L(\Omega) = \mathcal{Y}^L, \qquad \mathcal{Y} = \frac{1}{2}(Z_1 - Z_4)(\bar{Z}_1 + \bar{Z}_4) = r^2[i\sin\chi + \sinh(2\rho)], \tag{5.28}$$

where we have used (5.14). The bulk-to-boundary propagator reads

$$s^L(t_E) = c_L\left(\frac{1}{\cosh t_E}\right)^L, \qquad c_L = \ell_P^{9/2}\frac{2^L \pi \sqrt{k}}{2R^{9/2}}\frac{L+1}{L}\sqrt{2L+1}. \tag{5.29}$$

**The WZ action.** The WZ action is given by

$$\delta S_{WZ} = \frac{R_{AdS}^6}{(2\pi)^5 \ell_P^6} \frac{(2\pi)^3}{k} \int_{-\rho^{\max}}^{+\rho^{\max}} d\rho \, F_{WZ}\big|_{t_E=0}, \tag{5.30}$$

where

$$F_{WZ} = 2^8 \alpha^3 (1 - 2\alpha^2 \cosh(2\rho)) g^{r\beta} \partial_\beta Y_L(\Omega) s^L(t_E). \tag{5.31}$$

**Operator insertion.** Combining the two contributions, we find the operator $\hat{\mathcal{O}}_L$ evaluated on the classical solution $X_0^*$ is given by

$$\hat{\mathcal{O}}_L[X_0^*] = -\frac{R_{\text{AdS}}^6}{(2\pi)^5 \ell_P^6} \frac{(2\pi)^3}{k} \int_{-\rho^{\max}}^{+\rho^{\max}} d\rho \ (F_{\text{DBI}} - F_{\text{WZ}})|_{t_E=0} \ . \tag{5.32}$$

Plugging in the bulk-to-boundary propagator (5.29) and the spherical harmonics (5.28), we find more explicitly

$$F_{\text{DBI}} = 32 c_L (-1)^{L/2} L \alpha^{2L+2} \frac{2\alpha^2 \cosh(2\rho) - 1}{(\cosh t_E)^L} \left(4\alpha^2 - \cosh(2\rho)[1 + \tanh^2 t]\right) \tag{5.33}$$
$$\times (\sin\chi - i \sinh(2\rho))^L \, ,$$

$$F_{\text{WZ}} = 64 c_L (-1)^{L/2} L \alpha^{2L+2} \frac{2\alpha^2 \cosh(2\rho) - 1}{(\cosh t_E)^L} \left([2\alpha^2 - \cosh(2\rho)] \sin\chi - 2i\alpha^2 \sinh(2\rho)\right)$$
$$\times (\sin\chi - i \sinh(2\rho))^{L-1} \, .$$

To perform the orbit average, we shift $t_E \to t_E + \tau_0$ and $\chi \to \chi + \chi_0$ and integrate over $\tau_0$ and $\chi_0$. The structure constant is given by

$$C_{\mathcal{D}_M \mathcal{D}_M \mathcal{O}_L} = [(n_1 \cdot \bar{n}_2)(n_2 \cdot \bar{n}_1)]^{\frac{L}{2}-M} \left[\prod_{i=1}^{2} (n_i \cdot \bar{n}_3)(n_3 \cdot \bar{n}_i)\right]^{-\frac{L}{2}}$$
$$\times \int_{-\infty}^{\infty} d\tau_0 \int_0^{2\pi} \frac{d\chi_0}{2\pi} \hat{\mathcal{O}}_L[X_{\tau_0,\chi_0}^*], \tag{5.34}$$

where $\hat{\mathcal{O}}_J[X_{\tau_0,\chi_0}^*]$ is obtained by replacing $t_E$ and $\chi$ by $\tau_0$ and $\chi + \chi_0$ in $\hat{\mathcal{O}}_J[X_0^*]$ respectively. The details of the computations can be found in Appendix C , the final result is only non-vanishing for even $L$ and reads

$$C_{\mathcal{D}_M \mathcal{D}_M \mathcal{O}_L} = \left(\frac{\lambda}{2\pi^2}\right)^{1/4} \frac{\sqrt{2L+1}}{L} (1 + (-1)^L) \frac{(-1)^{\frac{L}{2}+1} 2^L \sqrt{\pi} \Gamma(\frac{L}{2}+1)}{\Gamma(\frac{L+3}{2})} (1 - 4\alpha^4)^{\frac{1}{2}(L-1)} \tag{5.35}$$
$$\times \left[(1 - 4\alpha^4) \, _2F_1\left(-\frac{1}{2}(L+1), -\frac{L}{2}; 1; \frac{4\alpha^4}{4\alpha^4-1}\right) \right.$$
$$\left. + 2\alpha^4 (L+1) \, _2F_1\left(-\frac{1}{2}(L-1), -\frac{L}{2}+1; 2; \frac{4\alpha^4}{4\alpha^4-1}\right)\right] \, .$$

## 5.2 Comparison with gauge theory

Although we do not expect a match, it is interesting to compare the results we obtained with the one from gauge theory. The structure constant for the BPS operators in the twisted translated frame has been computed in our previous paper [1] and we quote here

$$C_{\mathcal{D}_M \mathcal{D}_M \mathcal{O}_L}^{(\text{gauge})} = \sum_{p=-\frac{L}{2}}^{\frac{L}{2}} (-1)^p D_{M|L}^{(p)} =$$
$$\begin{cases} -\frac{(-1)^{\frac{L}{2}}}{\sqrt{L}} \left[P_{\frac{L}{2}}\left(-1 + 4\omega - 2\omega^2\right) + P_{\frac{L}{2}-1}\left(-1 + 4\omega - 2\omega^2\right)\right] & L : \text{even} \\ 0 & L : \text{odd} \end{cases} , \tag{5.36}$$

where $P_{\frac{L}{2}}$ is the Legendre polynomial. The parameters of the strong and weak coupling are related by

$$\frac{M}{N} \equiv \omega = \sqrt{1 - 4\alpha^4} - 4\alpha^4 \log\left(\frac{1 + \sqrt{1 - 4\alpha^4}}{2\alpha^2}\right), \qquad \frac{N}{k} \equiv \lambda. \tag{5.37}$$

The structure constant for fixed $L$ can be written down straightforwardly, for example

- $L = 2$

$$C^{(\text{gauge})}_{\mathcal{D}_M \mathcal{D}_M \mathcal{O}_2} = -\sqrt{2}\omega(\omega - 2),$$ 

(5.38)

$$C^{(\text{sugra})}_{\mathcal{D}_M \mathcal{D}_M \mathcal{O}_2} = \left(\frac{\lambda}{2\pi^2}\right)^{\frac{1}{4}} \frac{16\sqrt{5}}{3}(1 - 4\alpha^4)^{\frac{3}{2}}.$$

- $L = 4$

$$C^{(\text{gauge})}_{\mathcal{D}_M \mathcal{D}_M \mathcal{O}_4} = -\omega(\omega - 2)(3\omega^2 - 6\omega + 2),$$

(5.39)

$$C^{(\text{sugra})}_{\mathcal{D}_M \mathcal{D}_M \mathcal{O}_4} = -\left(\frac{\lambda}{2\pi^2}\right)^{\frac{1}{4}} \frac{128}{5}(1 - 4\alpha^4)^{\frac{3}{2}}(1 - 14\alpha^4).$$

- $L = 6$

$$C^{(\text{gauge})}_{\mathcal{D}_M \mathcal{D}_M \mathcal{O}_6} = -\sqrt{\frac{2}{3}}\omega(\omega - 2)(10\omega^4 - 40\omega^3 + 52\omega^2 - 24\omega + 3),$$

(5.40)

$$C^{(\text{sugra})}_{\mathcal{D}_M \mathcal{D}_M \mathcal{O}_6} = \left(\frac{\lambda}{2\pi^2}\right)^{\frac{1}{4}} \frac{2048\sqrt{13}}{105}(1 - 4\alpha^4)^{\frac{3}{2}}(1 - 36\alpha^4 + 198\alpha^8).$$

# 6 Conclusion and Discussion

## 6.1 Conclusion

In this paper, we revisited the holographic computation of two (non-maximal) giant gravitons and one single-trace BPS operator. We pointed out the incompleteness of the analysis in the literature. In particular we showed that the previous analyses missed two important effects; the orbit average and the boundary contribution coming from wave functions. For the case of $\mathcal{N} = 4$ SYM, we demonstrated that these effects are essential in reproducing the results computed in the gauge theory. We emphasize that this is a rather nontrivial match, since the final result shows a complicated dependence on the charge of the giant gravitons and is given by the Legendre polynomials or hypergeometric functions. For ABJM theory, our results make solid predictions on the structure constants at strong coupling, which can be compared with the integrability approach to be developed in the third paper [2].

If you are not aficionados of $\mathcal{N} = 4$ SYM and integrability, you might feel that all we did was to add yet another item to the already existing long list of precision tests of AdS/CFT (which we do not doubt anyway!). This is of course true to some extent, but let us emphasize that the fact that we succeeded in reproducing off-diagonal structure constants from the holographic computation is far from trivial: One might think that the semi-classical computation using the D-branes is sensitive only to the charges of order $N$ and cannot distinguish $O(1)$ differences. What we found in this paper is the contrary; the semi-classical computation *does* know the details of the heavy states if the computation is performed correctly. This "unreasonable effectiveness" of the semi-classical computation naturally brings us to the following question:

> *Is the method developed in this paper applicable to operators dual to black hole microstates? If so, how does the semiclassics distinguish different microstates?*

This is perhaps the most important but challenging future direction, and we will discuss it separately in the next subsection.

Here we list other future directions which are more integrability oriented, more technical, but perhaps more "low-hanging": In this paper, we focused on the giant gravitons, which are dual to sub-determinant operators. It would be interesting to extend our analysis to *dual giant gravitons*, which are dual to symmetric Schur polynomials. Another interesting future direction is to apply our method to *off-diagonal* heavy-heavy-light three-point functions of single-trace operators [20, 21]. Most of the works done for the heavy-heavy-light three-point functions focused on the diagonal three-point functions for which two heavy single-trace operators are identical up to conjugation. One of the few works which discussed the off-diagonal three-point functions is [34]. They analyzed off-diagonal three-point functions at weak coupling and pointed out that the results depend on the details of the two heavy operators and the semi-classical approximation based on the coherent states breaks down in some cases. However their analysis did not include the orbit average or the boundary terms from wave functions. It is plausible that the inclusion of these effects[21] resolve the discrepancy pointed out in [34]. If this is the case, that will open up a possibility to compute the *off-diagonal form factor* from semi-classical strings at strong coupling: As was demonstrated in [18], by taking a suitable limit of the diagonal heavy-heavy-light structure constants, one can read off the so-called diagonal form factors. Performing a similar analysis to the off-diagonal three-point function would give the off-diagonal form factors and would provide useful data to compare with the hexagon formalism for the three-point function [39]. In particular, the paper [40] computed off-diagonal heavy-heavy-light three-point functions at strong coupling using hexagons, and it would be interesting to reproduce their results from semi-classical strings. In [3], it was found that the contribution to three point functions from the open string attaching on $Z = 0$ brane is divergent. We now expect that the result will become finite after orbit average and taking into account the contribution from wave functions.

## 6.2 Application to black holes

We now discuss (and speculate on) the extent to which our method of computing holographic correlation functions of heavy operators applies to states dual to black holes. Before addressing this question, let us point out right away one important difference. The D-brane state discussed in this paper is expected to correspond to one particular operator in CFT (which in our case was a sub-determinant operator). By contrast a state dual to a black hole comes with exponentially large degeneracy ($\sim e^{N^2}$), as predicted by the Bekenstein-Hawking entropy. Therefore a conservative viewpoint is that whatever is computed in the semi-classical black hole background would correspond to an averaged result over such a large number of states. A closely related idea is that typical states as heavy as black holes exhibit a universal behavior dictated by the eigenstate thermalization [41] and the computation in the semi-classical black hole background captures that universal piece. In particular, the eigenstate thermalization predicts[22] the following answer for the matrix element of a light operator $\mathcal{O}$,

$$\langle E_m | \mathcal{O} | E_n \rangle = O_{\text{th}}(E_m)\delta_{mn} + e^{-S(\bar{E})/2} f_{\mathcal{O}}(\bar{E}, \Delta E) r_{mn}, \tag{6.1}$$

with

$$\bar{E} \equiv \frac{E_m + E_n}{2}, \qquad \Delta E \equiv \frac{E_m - E_n}{2}. \tag{6.2}$$

---

[21]One technical complication for analyzing these three-point functions is that, since the system under consideration is integrable, one needs to include the effects of higher conserved charges in the orbit average. In addition, the wave function needs to include coordinates dual to higher conserved charges. This latter problem can be solved by using Sklyanin's separation of variables [35] as was demonstrated in the context of the heavy-heavy-heavy three-point functions [36–38].

[22]Note, however, it is known that there are cases in which the formula gets modified. For instance, in 2d CFT, when the two states are in the same Verma module, off-diagonal elements are not exponentially suppressed ($e^{-S}$) but are power-law suppressed [42]. We thank Shouvik Datta for explaining this point.

Here $O_{\text{th}}(E)$ is the thermal expectation value of $\mathcal{O}$ for an ensemble with a mean energy $E$ and $f_{\mathcal{O}}(E,\delta)$ is some smooth function of $E$ and $\delta$ while $r_{mn}$ is a random variable with a unit variance. As can be seen in the formula, the off-diagonal element comes with a factor $e^{-S(\bar{E})/2}$ with $S$ being the thermal entropy that scales as $N^2$ for holographic states dual to black holes. Now, if one considers an average of (6.1), we would only see the diagonal part[23] and may conclude that the semi-classical computation in the black hole background would be ignorant of or insensitive to off-diagonal matrix elements.

However we have just seen in this paper that the semi-classical computation of D-brane states can capture off-diagonal parts once the computation is performed correctly. Given this success, it would be interesting to ask if there is a way to modify the naive semi-classical computation so that it becomes sensitive to the details of off-diagonal matrix elements. This is of course a difficult question both conceptually and technically. Therefore below we chart one possible path towards answering this question.

**Three-point functions of LLM backgrounds.**    Before studying black hole states, it would be useful to build our intuition and techniques using states that are heavy enough to deform the geometry but nevertheless are in the same "universality class" as the D-brane states discussed in this paper. The best candidates for such states are half-BPS states dual to the backgrounds constructed by Lin, Lunin and Maldacena [44]. Notable features of these geometries are that they do not have a horizon and the CFT duals of those states are known exactly [45, 46]. Therefore we can focus on technical aspects of how our method generalizes to states dual to nontrivial geometries without worrying about complications coming from conceptual aspects. In addition, we can test the holographic computation against the results in field theory.

At a practical level, we would need to find canonically conjugate variables associated with these backgrounds, in order to perform the orbit average and build the wave functions. For this purpose, the Crnkovic-Witten-Zuckerman quantization approach developed in [47,48] is likely to be useful.

**Matrix elements for Virasoro descendants.**    A possible next step would be to consider a black hole background but study a quantity with less conceptual difficulty. For instance, it would be interesting to try to compute the following ratio involving matrix elements of a light operator between a heavy primary state $|\text{primary}\rangle$ and its descendant $|\text{descandant}\rangle$ in AdS$_3$/CFT$_2$:

$$r \equiv \frac{\langle \text{descendant}|\mathcal{O}|\text{primary}\rangle}{\langle \text{primary}|\mathcal{O}|\text{primary}\rangle}\,. \tag{6.3}$$

On the CFT side, this ratio is completely fixed by the Virasoro symmetry [42]. On the gravity side, if $|\text{descendant}\rangle$ is obtained by the action of finitely many Virasoro generators on $|\text{primary}\rangle$, we expect that both states are described semi-classically by the BTZ black hole. The only difference between these two states is that the state $|\text{descendant}\rangle$ contains additional quanta of boundary gravitons as compared to $|\text{primary}\rangle$. The relevant "broken symmetry group" for performing the orbit average would be the asymptotic symmetry group (ASG), which acts on the Hilbert space of the boundary gravitons. Extrapolating the discussion in section 2, we can envisage a formula like

$$r \overset{\text{semi-classical}}{\sim} \int d[\text{ASG}]\, \Psi'^*_{\text{boundary graviton}}\, \mathcal{O}\, \Psi_{\text{boundary graviton}}\,. \tag{6.4}$$

---

[23]If we instead considers an average of a square of the matrix element, we would see a nontrivial contribution from the off-diagonal part since $|r_{mn}|^2 = 1$. Holographically, this is related to a configuration with Euclidean wormholes. See [43] for a recent discussion in the context of AdS$_3$/CFT$_2$.

Of course, this is just a speculation at the moment and the details need to be worked out.

Alternatively we can look at a matrix element between an eigenstate of the KdV charges and its small deformation using the KdV black holes constructed in [49]. On the CFT side, such a matrix element is also constrained by the Virasoro algebra. This is an interesting question also from the point of view of integrability, since the KdV black holes can be described by the classical spectral curve, much like semi-classical strings in $AdS_5 \times S^5$. It would be interesting to see whether one can extract quantum properties of these black holes from the semi-classical quantization[24] of the spectral curve [51].

**Microstates and horizon soft hair?**    We now turn to the most interesting question. Can we compute off-diagonal matrix elements of different black hole microstates using a semi-classical description in the bulk? In the context of AdS$_3$/CFT$_2$, this would correspond to computing the matrix elements of states in different Verma modules. Unlike the matrix elements of descendants, they depend on the details of the microscopic theory and contain dynamical information. A priori, it is not clear if the semi-classical description based on the BTZ black hole is capable of doing that. However, there is an interesting proposal by Hawking, Perry and Strominger [52, 53] which suggests that the soft hair at the horizon can distinguish microstates. Application of this idea to BTZ black holes was also discussed in the literature. At the time of writing this paper, it seems that no definitive conclusion has been made on the subject, but several interesting results came out of such studies. For instance, the paper [54–56] found the $U(1) \times U(1)$ Kac-Moody algebra as the asymptotic symmetry group at the horizon and proposed a description of microstates based on that algebra while the paper [57] wrote down a Schwarzian-like boundary action governing the reparametrization modes in the BTZ geometry. It would be interesting to push these ideas further and test them against CFT if possible. Ultimately, we would like to have a formula that generalizes (6.4) to the horizon symmetry group (HSG):

$$\langle \Psi_1 | \mathcal{O} | \Psi_2 \rangle \sim \int d[\text{HSG}] \, \Psi_1^* \mathcal{O} \Psi_2 \,. \tag{6.5}$$

Of course, it is not guaranteed that such a formula would exist. In fact, if it exists, it would predict some universal property of these off-diagonal matrix elements since the right hand side of (6.5) does not seem to know the microscopic detail of the theory. In this paper we remain agnostic about what the expectation is. However we want to emphasize that this is an important direction for the future, whatever the outcome will be.

**Superstrata and fuzzball.**    A related but slightly different direction is to compute the structure constants of two superstrata and a light supergravity mode in the D1-D5 system. The superstrata are BPS horizonless solutions in six-dimensional supergravity discussed in the context of the fuzzball program [58], which are conjectured to represent microstates of supersymmetric D1-D5 black holes with three charges[25] [60]. Unfortunately, the number of states given by the superstrata is not enough to fully account for the Bekenstein-Hawking entropy of the corresponding black holes [61], and it was argued in [62] that they can only describe *atypical* microstates. For the purpose of understanding quantum properties of typical black holes, this is certainly an undesired feature. However, the advantage is that one can perform a precision check of the holography: The superstrata (and three-charge black holes) are dual to large-charge 1/8-BPS operators in the dual CFT$_2$. Thanks to the non-renormalization theorem [10], the structure constants of two such operators and a chiral primary operator corresponding to

---

[24]There is also a possibility that we can express it using the separation of variables. See for instance [50].

[25]See [59] for a recent review.

a supergravity mode are protected. Therefore, one can directly compare the prediction from holography with the result from CFT$_2$. Such computations were performed already in the literature [63–66], building on earlier works [67–70]. However so far the analysis was limited to the diagonal three-point functions. It would be interesting to generalize such analyses to off-diagonal three-point functions and understand in detail to what extent the superstrata are atypical by making comparison with the eigenstate thermalization (6.1).

It is also worth mentioning that Skenderis and Taylor [67] pointed out that the fuzzball geometries without averaging do not correspond to eigenstates of the $R$-symmetry in the dual CFT; they instead correspond to superpositions of the eigenstates. This is more like the "converse" of what we found in this paper; namely the eigenstates of the R-symmetry correspond to superpositions—or more precisely an average—of classical (D-brane) solutions. It would be interesting to apply the idea of the orbit average to fuzzball solutions and try to compute correlation functions of $R$-symmetry eigenstates.

# Acknowledgements

We thank Alexandre Belin, Nikolay Bobev and Shouvik Datta for related discussions. We are in particular grateful to Shouvik Datta for explanations on the eigenstate thermalization hypothesis and its relation to AdS$_3$/CFT$_2$, and comments on the draft. We are also grateful to Francesco Aprile, James Drummond, Paul Heslop and Michele Santagata for useful comments on the first version of this paper and giving a detailed explanation on the single-particle basis. The work of SK was supported in part by DOE grant number DE-SC0009988. The work of PY and JBW is supported in part by the National Natural Science Foundation of China, Grant No. 11975164, 11935009, 12047502, 11947301, and Natural Science Foundation of Tianjin under Grant No. 20JCYBJC00910.

# A  Diagonal Structure Constants of Dual Giant Gravitons in $\mathcal{N} = 4$ SYM

In this appendix, we compute the diagonal structure constant of symmetric Schur polynomials dual to Giant Gravitons and a single-trace BPS operator in $\mathcal{N} = 4$ SYM at weak coupling. This generalizes the computation performed in section 3.2.

The only difference is that, instead of using the determinant (3.37) as a generating function, we need to use an inverse of a determinant

$$\mathcal{G}_j \equiv \frac{1}{\det\left[\mathbf{1} - t_j(Y_j \cdot \Phi)\right]}(x_j). \tag{A.1}$$

As is the case with the giant gravitons, the operator with a fixed charge $M$ can be obtained by an integral

$$\oint \frac{\mathrm{d}t_j}{2\pi i \, t_j^{1+M}} \mathcal{G}_j. \tag{A.2}$$

To proceed, we express the generating function in terms of integrals of *bosons*

$$\mathcal{G}_j = \int \mathrm{d}\bar{\phi}_j \mathrm{d}\phi_j \exp\left[-\bar{\phi}_j(\mathbf{1} - t_j Y_j \cdot \Phi)\phi_j\right]. \tag{A.3}$$

Following the derivation explained in the main text, we arrive at the expression

$$\langle \mathcal{G}_1 \mathcal{G}_2 \mathcal{O}_L \rangle = \frac{1}{Z} \int d\rho \, \langle \mathcal{O}_L^S \rangle_\phi \exp \left[ \underbrace{-\frac{2N}{g^2} \rho_{12}\rho_{21} - N \log\left(1 - 4t_1 t_2 d_{12} \rho_{12}\rho_{21}\right)}_{\equiv S_{\text{eff}}} \right]. \tag{A.4}$$

Here $\langle \mathcal{O}_L^S \rangle_\phi$ is obtained by replacing $\mathcal{O}_L$ with

$$\mathcal{O}_L^S(x_3) = \text{tr}\left((Y_3 \cdot S)^L\right), \qquad S^I = \frac{g_{\text{YM}}^2}{8\pi^2} \sum_{k=1,2} \frac{t_k Y_k^I \phi_k \bar{\phi}_k}{|x_{k3}|^2}. \tag{A.5}$$

The Wick contraction of bosons is given by

$$\left\langle \bar{\phi}_i^a \phi_{j,b} \right\rangle = \delta_b^a (\Sigma^{-1})_{ij}, \tag{A.6}$$

where

$$\Sigma = \begin{pmatrix} 1 & 2\hat{\rho}_{12} \\ 2\hat{\rho}_{21} & 1 \end{pmatrix}, \tag{A.7}$$

and $\hat{\rho}_{ij} = \sqrt{t_i t_j d_{ij}} \rho_{ij}$.

The saddle point for $\rho$ is given by

$$\rho_{12}^* \rho_{21}^* = \frac{1}{4t_1 t_2 d_{12}} - \frac{g^2}{2}, \tag{A.8}$$

while the action at the saddle-point is given by

$$S_{\text{eff}} = -N\left(-1 + \frac{1}{2g^2 t_1 t_2 d_{12}} + \log(2g^2 t_1 t_2 d_{12})\right). \tag{A.9}$$

Using this effective action, we can compute the saddle point of $t_{1,2}$. As a result, we obtain

$$t_1^* t_2^* = \frac{1}{2g^2 \cosh^2 \eta \, d_{12}}, \tag{A.10}$$

where $\eta$ is given in terms of the charge of the dual giant graviton by

$$\frac{M}{N} \equiv \sinh^2 \eta. \tag{A.11}$$

We then get

$$\langle \mathcal{O}_L^S \rangle_\phi = (2g^2)^{\frac{L}{2}} \left(\frac{d_{13} d_{23}}{d_{12}}\right)^{\frac{L}{2}} \oint_{|y|=1} \frac{dy}{2\pi i y} \text{Tr}\left[\hat{\mathcal{T}}^L\right], \tag{A.12}$$

with

$$\hat{\mathcal{T}} \equiv \text{diag}\left(y, y^{-1}\right) \cdot \begin{pmatrix} \cosh\eta & \sinh\eta \\ \sinh\eta & \cosh\eta \end{pmatrix}. \tag{A.13}$$

Here again, the integral of $y$ comes from the integral of the ratio $t_1/t_2$, which is not fixed by the saddle-point equation (A.10).

Using the same rewriting as before, we obtain

$$\oint_{|y|=1} \frac{\mathrm{d}y}{2\pi i y} \mathrm{Tr}\big[\hat{\mathcal{T}}^L\big] = 2 \oint_{|s|=\epsilon\ll 1} \frac{\mathrm{d}s}{2\pi i s^{1+L}} \oint_{|y|=1} \frac{\mathrm{d}y}{2\pi i y} \frac{1-\frac{s}{2}(y+\frac{1}{y})\cosh\eta}{1-s(y+\frac{1}{y})\cosh\eta+s^2}. \tag{A.14}$$

Performring the integral of $y$ by computing the residue, we get

$$\oint_{|y|=1} \frac{\mathrm{d}y}{2\pi i y} \mathrm{Tr}\big[\hat{\mathcal{T}}^L\big] = \oint_{|s|=\epsilon\ll 1} \frac{\mathrm{d}s}{2\pi i s^{1+L}} \left[ 1 + \frac{1-s^2}{\sqrt{1-2s^2\cosh 2\eta+s^4}} \right]. \tag{A.15}$$

Combining everything, we arrive at the final result

$$\big\langle \mathcal{O}_L^S \big\rangle_\phi = (2g^2)^{\frac{L}{2}} \left( \frac{d_{13}d_{23}}{d_{12}} \right)^{\frac{L}{2}} \frac{1+(-1)^L}{2} \left( P_{\frac{L}{2}}(\cosh 2\eta) - P_{\frac{L}{2}-1}(\cos 2\eta) \right), \tag{A.16}$$

which leads to the following result for the structure constant

$$C_{\mathcal{D}_M\mathcal{D}_M\mathcal{O}_L} = \frac{1+(-1)^L}{2\sqrt{L}} \left( P_{\frac{L}{2}}(\cosh 2\eta) - P_{\frac{L}{2}-1}(\cosh 2\eta) \right). \tag{A.17}$$

It is an interesting future problem to reproduce (A.17) from the holographic computation using dual giant gravitons at strong coupling.

# B  Strong Coupling Computation in $\mathcal{N}=4$ SYM

In this appendix, we give more details for the strong coupling computation of $\mathcal{N}=4$ SYM theory. We will derive (3.30) for the diagonal structure constant. For the non-diagonal case, we give a simple expression which allows us to compute the structure constant straightforwardly.

## B.1  Diagonal structure constant

Let us define

$$\bar{F}_{\mathrm{DBI}} = \frac{1}{2\pi} \int_0^{2\pi} F_{\mathrm{DBI}}(\tau_0, \phi_0)\mathrm{d}\phi_0, \qquad \bar{F}_{\mathrm{WZ}} = \frac{1}{2\pi} \int_0^{2\pi} F_{\mathrm{WZ}}(\tau_0, \phi_0)\mathrm{d}\phi_0. \tag{B.1}$$

Using (3.28), we can compute the integral over $\phi_0$, $\chi_3$ and $\chi_1$. The results for the DBI and WZ actions are given by

$$\int_0^{\pi/2} \mathrm{d}\chi_1 \int_0^{2\pi} \mathrm{d}\chi_3 \bar{F}_{\mathrm{DBI}} = \frac{\sqrt{L(L+1)}}{2N} \frac{\cos(2\theta_0)+\tanh^2\tau_0}{(\cosh\tau_0)^L} A_L(\theta_0), \tag{B.2}$$

$$\int_0^{\pi/2} \mathrm{d}\chi_1 \int_0^{2\pi} \mathrm{d}\chi_3 \bar{F}_{\mathrm{WZ}} = \frac{\sqrt{L(L+1)}}{2LN} \frac{\sin(2\theta_0)\partial_{\theta_0}A_L(\theta_0)}{(\cosh\tau_0)^L},$$

where $A_L(\theta_0)$ is given by

$$\frac{A_L(\theta_0)}{2\pi} = \frac{1+(-1)^L}{2} \frac{L!}{2^L} \sum_{n=0}^{L/2} \frac{(-1)^{\frac{L}{2}-n}}{(n!)^2[(\frac{L}{2}-n)!]^2} \frac{(\sin^2\theta_0)^n(\cos^2\theta_0)^{\frac{L}{2}-n}}{\frac{L}{2}-n+1}. \tag{B.3}$$

As a next step, we compute the $\tau_0$ integral. we find

$$C_{\mathcal{D}_M\mathcal{D}_M\mathcal{O}_L} = \delta S_{\mathrm{DBI}} + \delta S_{\mathrm{WZ}}, \tag{B.4}$$

where

$$\delta S_{\text{DBI}} = -\frac{\sqrt{L}(L+1)}{2}\cos^2\theta_0(2\cos^2\theta_0\, t_L - t_{L+2})\frac{A_L(\theta_0)}{2\pi}\,, \tag{B.5}$$

$$\delta S_{\text{WZ}} = \frac{L+1}{2\sqrt{L}}t_L\cos^2\theta_0\sin(2\theta_0)\frac{\partial_{\theta_0}A_L(\theta_0)}{2\pi}\,,$$

with

$$t_L = \frac{2^{L+1}}{L}\frac{[(L/2)!]^2}{L!}\,. \tag{B.6}$$

Plugging (B.3) into (B.5) and after some algebra, we find

$$C_{\mathcal{D}_M\mathcal{D}_M\mathcal{O}_L} = -\frac{1+(-1)^L}{\sqrt{L}}(-1)^{L/2}(\cos\theta_0)^L\, {}_2F_1\left(1-\tfrac{L}{2},-\tfrac{L}{2};1,-\tan^2\theta_0\right) \tag{B.7}$$

$$= -\frac{1+(-1)^L}{\sqrt{L}}(-1)^{L/2}\, {}_2F_1\left(-\tfrac{L}{2},\tfrac{L}{2};1,\sin^2\theta_0\right)\,.$$

Using the identity of the hypergeometric function

$${}_2F_1\left(-\tfrac{L}{2},\tfrac{L}{2};1,\sin^2\theta_0\right) = \frac{1}{2}\left[{}_2F_1\left(1-\tfrac{L}{2},\tfrac{L}{2};1,\sin^2\theta_0\right) + {}_2F_1\left(-\tfrac{L}{2},1+\tfrac{L}{2};1,\sin^2\theta_0\right)\right]\,, \tag{B.8}$$

and the relation between hypergeometric function and the Legendre polynomial

$${}_2F_1\left(1-\tfrac{L}{2},\tfrac{L}{2};1,\sin^2\theta_0\right) = P_{\frac{L}{2}-1}(\cos(2\theta_0))\,, \tag{B.9}$$

$${}_2F_1\left(-\tfrac{L}{2},1+\tfrac{L}{2};1,\sin^2\theta_0\right) = P_{\frac{L}{2}}(\cos(2\theta_0))\,,$$

we find

$$C_{\mathcal{D}_M\mathcal{D}_M\mathcal{O}_L} = -\frac{1+(-1)^L}{2\sqrt{L}}(-1)^{L/2}\left(P_{\frac{L}{2}}(\cos(2\theta_0)) + P_{\frac{L}{2}-1}(\cos(2\theta_0))\right)\,. \tag{B.10}$$

## B.2 Off-diagonal structure constant

As explained in the main text, for the off-diagonal structure constant, we need to take into boundary terms coming from wave function that comes from the wave functions. For two giant gravitons with charges $M_1$ and $M_2$ such that $M_1 - M_2 = k$, we define the following quantities

$$\bar{F}_{\text{DBI}}^{(k)} = \frac{1}{2\pi}\int_0^{2\pi} F_{\text{BDI}}(\tau_0,\phi_0)e^{ik\phi_0}e^{k\tau_0}\mathrm{d}\phi_0\,, \tag{B.11}$$

$$\bar{F}_{\text{WZ}}^{(k)} = \frac{1}{2\pi}\int_0^{2\pi} F_{\text{WZ}}(\tau_0,\phi_0)e^{ik\phi_0}e^{k\tau_0}\mathrm{d}\phi_0\,.$$

Notice that there are *two* phase factors in the asymmetric case, the red colored one comes from the $S^5$ part while the blue colored one comes from the AdS part.

The off-diagonal structure constant is given by

$$C_{\mathcal{D}_{M+k}\mathcal{D}_M\mathcal{O}_L}^{(k)} = -\frac{N}{2\pi}\cos^2\theta_0\int_{-\infty}^{\infty}\mathrm{d}\tau_0\int_0^{2\pi}\mathrm{d}\chi_3\int_0^{\pi/2}\mathrm{d}\chi_1\,\bar{F}_{\text{full}}^{(k)}\,, \tag{B.12}$$

where

$$\bar{F}_{\text{full}}^{(k)} = \bar{F}_{\text{DBI}}^{(k)} - \bar{F}_{\text{WZ}}^{(k)}. \tag{B.13}$$

For integer $L$, the integrals over $\phi_0$ and $\tau_0$ can be computed separately. We therefore introduce two integrals

$$A_{L,k}(c) = \frac{1}{2\pi} \int_0^{2\pi} (\cos\phi - c)^L e^{ik\phi} \, d\phi \,, \qquad B_{L,k} = \int_{-\infty}^{\infty} \frac{e^{k\tau_0}}{(\cosh\tau_0)^L} d\tau_0 \,. \tag{B.14}$$

They can be computed analytically at any positive integer values of $L, k$ with $L > k$:

$$A_{L,k}(c) = \frac{1}{2^L} C_{L+k}^{-L}(c) \,, \tag{B.15}$$

$$B_{L,k} = 2^L \left( \frac{{}_2F_1(L, \frac{L-k}{2}, \frac{L-k}{2}+1, -1)}{L-k} + \frac{{}_2F_1(L, \frac{L+k}{2}, \frac{L+k}{2}+1, -1)}{L+k} \right),$$

where $C_n^\alpha(x)$ is the Gegenbauer polynomial. Let us define the variable

$$\zeta = -i \cot\theta_0 \cos\chi_1 \sin\chi_3 \,. \tag{B.16}$$

We can write $F_{\text{full}}$ as

$$F_{\text{full}} = -\frac{\sqrt{L}(L+1)}{2N} \frac{(\sin\theta_0)^L \sin(2\chi_1)}{(\cosh\tau_0)^L} (\cos\phi - \zeta)^{L-1} \left( \frac{\cos\phi - \zeta}{(\cosh\tau_0)^2} + 2\zeta \right). \tag{B.17}$$

Therefore we find that

$$\bar{F}_{\text{bulk}}^{(k)} = -\frac{\sqrt{L}(L+1)}{2N} (\sin\theta_0)^L \sin(2\chi_1) \left[ A_{L,k}(\zeta) B_{L+2,k} + 2\zeta A_{L-1,k}(\zeta) B_{L,k} \right]. \tag{B.18}$$

For fixed $L$ and $k$, the above quantity can be computed straightforwardly. We consider two examples.

**Next-to-extremal**   Taking $k = L - 2$, we have

$$A_{L,L-2}(\zeta) B_{L+2,L-2} + 2\zeta A_{L-1,L-2}(\zeta) B_{L,L-2} = -\frac{8}{L+1}\zeta^2 + \frac{2}{L+1} \,, \tag{B.19}$$

and we have

$$\bar{F}_{\text{full}}^{(k)} = \frac{\sqrt{k+2}}{N} (\sin\theta)^{k+2} \sin(2\chi_1)(4\zeta^2 - 1) \,. \tag{B.20}$$

Plugging in the explicit forms of $\zeta$, the rest of the integrals over $\chi_1$ and $\chi_3$ can be computed straightforwardly, yielding

$$C_{\mathcal{D}_{M+k}\mathcal{D}_M\mathcal{O}_{k+2}} = \sqrt{k+2}(\cos\theta_0)^2 (\sin\theta_0)^k \,. \tag{B.21}$$

**Next-to-next-to-extremal**   Taking $k = L - 4$, we have

$$\bar{F}_{\text{full}}^{(k)} = \frac{\sqrt{k+4}}{N} (\sin\theta_0)^{k+4} \sin(2\chi_1) \left[ 4(k+1)\zeta^4 - 2(k-1)\zeta^2 - 1 \right]. \tag{B.22}$$

Again it is straightforward to evaluate the rest of the integrals and we find

$$C_{\mathcal{D}_{M+k}\mathcal{D}_M\mathcal{O}_{k+4}} = -\frac{\sqrt{k+4}}{2} (\cos\theta_0)^2 (\sin\theta_0)^k \left[ (\cos\theta_0)^2(k+3) - 2 \right], \tag{B.23}$$

both cases are in perfect agreement with the result from weak coupling computation.

# C  Strong Coupling Computation in ABJM

In this appendix, we present more details for the computation of the result (5.35). We first compute the orbit averaging over $\chi_0$

$$\bar{F}_{\text{DBI}} = \frac{1}{2\pi} \int_0^{2\pi} F_{\text{DBI}}(\chi + \chi_0) \mathrm{d}\chi_0 \,, \qquad \bar{F}_{\text{WZ}} = \frac{1}{2\pi} \int_0^{2\pi} F_{\text{WZ}}(\chi + \chi_0) \mathrm{d}\chi_0 \,. \tag{C.1}$$

This leads to

$$\bar{F}_{\text{DBI}} = 32 c_L (-1)^{L/2} L \alpha^{2L+2} \frac{2\alpha^2 \cosh(2\rho) - 1}{\cosh^L t} \left(4\alpha^2 - \cosh(2\rho)[1 + \tanh^2 t]\right) A_L \,, \tag{C.2}$$

$$\bar{F}_{\text{WZ}} = 64 c_L (-1)^{L/2} L \alpha^{2L+2} \frac{2\alpha^2 \cosh(2\rho) - 1}{\cosh^L t} \widetilde{A}_{L-1}(\rho) \,,$$

where

$$A_n = \frac{1}{2\pi} \int_0^{2\pi} (\sin(\chi + \chi_0) - i \sinh(2\rho))^n \mathrm{d}\chi_0 \,, \tag{C.3}$$

$$\widetilde{A}_n = [2\alpha^2 - \cosh(2\rho)] B_n(\rho) - 2i\alpha^2 \sinh(2\rho) A_n(\rho) \,,$$

and

$$B_n(\rho) = \frac{1}{2\pi} \int_0^{2\pi} \sin(\chi + \chi_0)(\sin(\chi + \chi_0) - i \sinh(2\rho))^n \mathrm{d}\chi_0 \,. \tag{C.4}$$

These integrals can be computed analytically, leading to

$$A_n(\rho) = n! \sum_{m=0}^{[n/2]} \frac{(-i)^{n-2m}}{4^m (n-2m)!(m!)^2} \left[\sinh(2\rho)\right]^{n-2m} \,, \tag{C.5}$$

$$B_n(\rho) = n! \sum_{m=1}^{[\frac{n+1}{2}]} \frac{(2m)(-i)^{n+1-2m}}{4^m (n+1-2m)!(m!)^2} \left[\sinh(2\rho)\right]^{n+1-2m} \,.$$

After performing the orbit average of $\chi_0$, $\chi$ drops out and the orbit average over $\tau_0$ integrals of $\bar{F}_{\text{DBI}}$ and $\bar{F}_{\text{WZ}}$ can be computed readily. Performing the integral over $\tau_0$, we find

$$\int_{-\infty}^{\infty} (\bar{F}_{\text{DBI}} - \bar{F}_{\text{WZ}}) \mathrm{d}\tau_0 = -c_L \frac{16\sqrt{\pi}(-1)^{L/2} L \Gamma(\frac{L}{2})}{\Gamma(\frac{L+3}{2})} \alpha^{2L+2} (1 - 2\alpha^2 \cosh(2\rho)) \cosh(2\rho) \tag{C.6}$$
$$\times G_L[\sinh(2\rho)] \,,$$

where

$$G_L[\sinh(2\rho)] = L B_{L-1}(\rho) + i(L+2) \sinh(2\rho) A_{L-1}(\rho) \tag{C.7}$$

is a polynomial of $\sinh(2\rho)$.

Finally we perform the integral over $\rho$, which is slightly involved, but we manage to find a closed form expression

$$C_{\mathcal{D}_M \mathcal{D}_M \mathcal{O}_L} = (-1)^{\frac{L}{2}+1} 2^L \frac{R^6}{(2\pi)^5 \ell_P^6} \frac{(2\pi)^3}{k} \int_{-\rho_{\max}}^{\rho_{\max}} \mathrm{d}\rho \int_{-\infty}^{\infty} (\bar{F}_{\text{DBI}} - \bar{F}_{\text{WZ}}) \mathrm{d}\tau_0 \tag{C.8}$$

$$= (-1)^{\frac{L}{2}+1} 2^L \frac{R_{\text{AdS}}^6}{(2\pi)^5 \ell_P^6} \frac{(2\pi)^3}{k} c_L (1 + (-1)^L) \frac{8\sqrt{\pi} \Gamma(\frac{L}{2}+1)}{(L+1) 2^L \Gamma(\frac{L+3}{2})} (1 - 4\alpha^4)^{\frac{1}{2}(L-1)}$$

$$\times \left[(1 - 4\alpha^4) \, _2F_1\left(-\frac{L+1}{2}, -\frac{L}{2}; 1; \frac{4\alpha^4}{4\alpha^4 - 1}\right) + 2\alpha^4 (L+1) \, _2F_1\left(-\frac{L-1}{2}, -\frac{L}{2}+1; 2; \frac{4\alpha^4}{4\alpha^4 - 1}\right)\right] \,.$$

The prefactor can be written as

$$\frac{R_{\text{AdS}}^6}{(2\pi)^5 \ell_P^6} \frac{(2\pi)^3}{k} c_L = \left(\frac{\lambda}{2\pi^2}\right)^{1/4} \frac{\sqrt{2L+1}(L+1)}{8L} 2^L .$$ (C.9)

Plugging this into (C.8), we obtain (5.35) in the main text.

# D  Beyond Twisted Translated Frame

The holographic computations in section 5 can be generalized beyond the twisted translated frame. As an illustration, we take the polarization vectors of the single-trace operator to be

$$n_3 = \frac{1}{\sqrt{2(\eta^2+1)}}(1, \eta, -1, \eta), \qquad \bar{n}_3 = \frac{1}{\sqrt{2(\eta^2+1)}}(1, -\eta, 1, \eta),$$ (D.1)

with real $\eta$[26]. The polarization vectors of the giant gravitons are unchanged. With this choice, the $R$-symmetry cross ratio becomes $\xi = \eta^2$. Now we can obtain $D_{M|L}^{(p)}$ from by reading off the coefficients of different powers of $\eta$ from the the structure constants $C_{\mathcal{D}_M \mathcal{D}_M \mathcal{O}_L}$. The spherical harmonics now takes the following form

$$Y_\eta = \frac{1}{2(\eta^2+1)}(Z_1 + \eta Z_2 - Z_3 + \eta Z_4)(\bar{Z}_1 - \eta \bar{Z}_2 + \bar{Z}_3 + \eta \bar{Z}_4).$$ (D.2)

The holographic computation is similar to what we did in section 5. Therefore we only give the final results for $C_{\mathcal{D}_M \mathcal{D}_M \mathcal{O}_L}$ and $D_{M|L}^{(p)}$. As a consistency check, we also compute combination $\sum_{p=-\frac{L}{2}}^{\frac{L}{2}} (-1)^p D_{M|L}^{(p)}$ and compare with the results in subsection 5.2.

- $L = 1$.

$$C_{\mathcal{D}_M \mathcal{D}_M \mathcal{O}_1} = \left(\frac{\lambda}{2\pi^2}\right)^{\frac{1}{4}} \sqrt{3}\pi \left(\frac{1}{\eta} + \eta\right) \left(\sqrt{1-4\alpha^4} - 4\alpha^4 \operatorname{arcsech}\left(2\alpha^2\right)\right),$$ (D.3)

from which we can read off

$$D_{M|1}^{(-1/2)} = D_{M|1}^{(1/2)} = \left(\frac{\lambda}{2\pi^2}\right)^{\frac{1}{4}} \sqrt{3}\pi \left(\sqrt{1-4\alpha^4} - 4\alpha^4 \operatorname{arcsech}\left(2\alpha^2\right)\right),$$ (D.4)

and find that

$$\sum_{p=-\frac{1}{2}}^{\frac{1}{2}} (-1)^p D_{M|1}^{(p)} = 0.$$ (D.5)

- $L = 2$.

$$C_{\mathcal{D}_M \mathcal{D}_M \mathcal{O}_2} = -\left(\frac{\lambda}{2\pi^2}\right)^{\frac{1}{4}} \frac{8\sqrt{5}}{3\eta^2} \Big( \sqrt{1-4\alpha^4} \left(4\alpha^4 \left(5\eta^4 + 12\eta^2 + 5\right) + \eta^4 + 1\right)$$ (D.6)
$$- 24\alpha^4 \left(\eta^2 + 1\right)^2 \operatorname{arcsech}\left(2\alpha^2\right)\Big),$$

---

[26]Without loss of generality, we can take $\eta$ to be positive.

from which we can read off

$$
\begin{aligned}
D_{M|2}^{(-1)} = D_{M|2}^{(1)} \\
= -\left(\frac{\lambda}{2\pi^2}\right)^{\frac{1}{4}}\left(\frac{160}{3}\sqrt{5(1-4\alpha^4)}\alpha^4 + \frac{8}{3}\sqrt{5(1-4\alpha^4)} - 64\sqrt{5}\alpha^4\operatorname{arcsech}\left(2\alpha^2\right)\right),
\end{aligned}
$$
(D.7)

$$
D_{M|2}^{(0)} = -\left(\frac{\lambda}{2\pi^2}\right)^{\frac{1}{4}}128\sqrt{5}\alpha^4\left(\sqrt{(1-4\alpha^4)} - \operatorname{arcsech}\left(2\alpha^2\right)\right),
$$

and check that

$$
\sum_{p=-1}^{1}(-1)^p D_{M|2}^{(p)} = \left(\frac{\lambda}{2\pi^2}\right)^{\frac{1}{4}}\frac{16\sqrt{5}}{3}(1-4\alpha^4)^{\frac{3}{2}}.
$$
(D.8)

- $L = 3$.

$$
\begin{aligned}
\mathcal{C}_{\mathcal{D}_M\mathcal{D}_M\mathcal{O}_3} = -\left(\frac{\lambda}{2\pi^2}\right)^{\frac{1}{4}}\frac{\sqrt{7}\pi}{\eta^3}\left(\eta^2+1\right)\left(-\sqrt{1-4\alpha^4}\left(2\alpha^4\left(52\eta^4+137\eta^2+52\right)\right.\right. \quad\text{(D.9)}\\
\left. + \eta^4 - \eta^2 + 1\right) + 72\alpha^4\left(\alpha^4\left(2\eta^4+7\eta^2+2\right)\right.\\
\left.\left. + \left(\eta^2+1\right)^2\right)\operatorname{arcsech}\left(2\alpha^2\right)\right),
\end{aligned}
$$

from which we can read off

$$
D_{M|3}^{(-\frac{3}{2})} = D_{M|3}^{(\frac{3}{2})} \quad\text{(D.10)}
$$

$$
= -\left(\frac{\lambda}{2\pi^2}\right)^{\frac{1}{4}}\sqrt{7}\pi\left(72\alpha^4\left(2\alpha^4+1\right)\operatorname{arcsech}\left(2\alpha^2\right) - \sqrt{1-4\alpha^4}\left(104\alpha^4+1\right)\right),
$$

$$
D_{M|3}^{(-\frac{1}{2})} = D_{M|3}^{(\frac{1}{2})} = \left(\frac{\lambda}{2\pi^2}\right)^{\frac{1}{4}}54\sqrt{7}\pi\alpha^4\left(4\left(3\alpha^4+1\right)\operatorname{arcsech}\left(2\alpha^2\right) - 7\sqrt{1-4\alpha^4}\right),
$$

and check that

$$
\sum_{p=-\frac{3}{2}}^{\frac{3}{2}}(-1)^p D_{M|3}^{(p)} = 0.
$$

We can see that the combination $\sum_{p=-\frac{L}{2}}^{\frac{L}{2}}(-1)^p D_{M|L}^{(p)}$ indeed reproduce exactly the results in the twisted translated frame given in subsection 5.2 for $L = 1, 2, 3$. Generalization to higher $L$ is straightforward.

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
