# Peer review of "D-branes and Orbit Average"

_SciPost Physics, doi:SciPost Phys. 12, 055 (2022)_

## Round 3 · Referee Report · Anonymous · 2021-10-25

Strengths

1. The problem is clearly stated and the background leading to the resolution is reviewed with examples
2. Provides an improved framework for further Heavy-Heavy-Light 3-point function computations in the future
3. Some of the effects noticed might also have applications to the holographic understanding of black hole states
4. Very well written with minimal typos

Weaknesses

1. Despite presenting a more consistent framework than previous attempts, this work does not actually resolve the ambiguities arising in extremal 3-point function computations. However, the authors do state the different aspects of the problem in better detail than had been done previously, which might lead to future progress.

Report

In this work, the authors revisit the computation of 3-point functions involving two (sub)determinant operators and a single-trace BPS operator, in N=4 SYM and the ABJM theory. In the extremal case, past work had produced an unexpected mismatch between the gravity and gauge sides of the AdS/CFT correspondence, which was resolved via a regularisation method. Since those initial attempts, the understanding of how to work with semiclassical correlators has improved and the authors apply these more modern techniques to provide a more rigorous computation of such correlators.
In particular, they take into account an average over classical solutions with the same charges, and also the fact that the two classical ("heavy") operators differ by O(1) terms. The fact that these O(1) terms do contribute and enter the semiclassical computation is perhaps one of the main intriguing outcomes of this work.
Although the current computation does not resolve the extremal 3-point function computation (in the sense that to obtain a matching one still requires analytical continuation from the non-extremal case, which comes with its own assumptions), it certainly shines additional light onto the problem and the techniques used here are likely to lead to further progress in the future, both in other 3-point function settings and (more speculatively) in computations involving states dual to black holes. The current techniques also lead to improvements in non-extremal computations with impressive matching between the gauge and gravity sides.
In all, the current work has addressed a past stumbling block (which had previously been resolved but perhaps in a slightly ad-hoc way) using newly-developed techniques, and provided, if not a breakthrough, certainly a well-defined framework in which to tackle such computations in the future. There is clear potential for follow-up work and the authors detail several such possibilities.
The paper is written in a clear way, cites the appropriate literature and the computations are presented in sufficient detail to allow one to reproduce them. A simple example explains the main physical intuition which is then applied to the problem under study. The paper concludes with a detailed discussion of possible applications, both conceptual and computational.
I thus consider this work appropriate for publication in SciPost Physics.

---

## Round 3 · Referee Report · Anonymous · 2021-11-4

Strengths

(1) A careful treatment of the semiclassical computation of amplitudes for interaction of giant graviton branes with perturbative gravitons.
(2) A critical review of the relevant literature on these semiclassical computations.
(3) comparison of present work with previous approaches.

Weaknesses

(1) No significant weaknesses.

Report

I am happy to recommend publication in this journal.

---

## Editorial Decision

published